# Power Spectra of Random Heterogeneities in the Solid Earth

Haruo Sato[1]

[1]Geophysics, Science, Tohoku University, Aoba-ku, Sendai-shi, Miyagi-ken, *980-8578*, Japan

**Correspondence:** Haruo Sato (haruo.sato.e2@tohoku.ac.jp)

**Abstract.** Recent seismological observations focusing on the collapse of an impulsive wavelet revealed the existence of small-scale random heterogeneities in the earth medium. The radiative transfer theory (RTT) is often used for the study of the propagation and scattering of wavelet intensities, the mean square amplitude envelopes through random media. For the statistical characterization of the power spectral density function (PSDF) of the random fractional fluctuation of velocity inhomogeneities in a 3D space, we use an isotropic von Kármán type characterized by three parameters: the root mean square (RMS) fractional velocity fluctuation, the characteristic length, and the order of the modified Bessel function of the second kind, which lead to the power-law decay of PSDF at wavenumbers higher than the corner. We compile reported statistical parameters of the lithosphere and the mantle based on various types of measurements for a wide range of wavenumbers: photo scan data of rock samples, acoustic well log data, and envelope analyses of cross-hole experiment seismograms, regional seismograms and tele-seismic waves based on the RTT. Reported exponents of wavenumber are distributed between –3 and –4, where many of them are close to –3. Reported RMS fractional fluctuations are of the order of 0.01∼0.1 in the crust and the upper mantle. Reported characteristic lengths distribute very widely, however, each one seems to be restricted by the dimension of the measurement system or the sample length. In order to grasp the spectral characteristics, eliminating strong heterogeneity data and the lower mantle data, we have plotted all the reported PSDFs of the crust and the upper mantle against wavenumber for a wide range $10^{-3} \sim 10^{8}$ km$^{-1}$. We find that the spectral envelope of those PSDFs is well approximated by the –3rd power of wavenumber. It suggests that the earth medium randomness has a broad spectrum. In theory, we need to re-examine the applicable range of the Born approximation in the RTT when the wavenumber of a wavelet is much higher than the corner. In observation, we will have to measure carefully the PSDF on both sides of the corner. We may consider the obtained power-law decay spectral envelope as a reference for studying the regional differences. It is interesting to study what kinds of geophysical processes created the observed power-law spectral envelope in different scales and in different geological environments in the solid earth medium.

File name: Sato_SE_20190112

Print date: January 12, 2019

# 1 Introduction

The first image of the solid earth is composed of spherical shells, for example, PREM (Dziewonski and Anderson, 1981). As seismic networks were deployed on the regional scale and worldwide, the velocity tomography based on the ray tracing method revealed 3D heterogeneous structure in various scales; however, spatial variations of the resultant velocity structure are essentially smooth compared with seismic wavelengths. Aki and Chouet (1975) first put a focus on long lasting coda waves of small earthquakes and interpreted them as scattered waves by small-scale random heterogeneities. They proposed to measure the scattering coefficient $g$, the scattering power per unit volume as a measure of medium heterogeneity. They analyzed the mean square (MS) amplitude time trace of coda waves as an incoherent sum of scattered waves' power by using the Born approximation (e.g. Chernov, 1960), which is a simplified version of the radiative transfer theory (RTT). There have been many measurements of the total scattering coefficient $g_{iso}$ supposing isotropic scattering (e.g. Sato, 1977a) in various seismo-tectonic environments. The total scattering coefficient of S-waves is reported to be of the order of $10^{-2}$ km$^{-1}$ for $1 \sim 20$ Hz in the lithosphere, and it marks a higher value beneath active volcanoes (e.g. Sato et al., 2012; Yoshimoto and Jin, 2008). There were precise measurements of regional variations in $g_{iso}$ as Carcolé and Sato (2010) in Japan and Eulenfeld and Wegler (2017) in US. Hock et al. (2004) analyzed medium heterogeneity in Europe from the analyses of tele-seismic waves using the modified energy flux model (Korn, 1993). There were also measurements of the anisotropic scattering coefficient from the analysis of S coda envelopes (e.g. Jing et al., 2014; Zeng, 2017).

Aki and Chouet (1975) derived the angular dependence of the scattering coefficient of scalar waves from the power spectral density function (PSDF) of the fractional velocity fluctuation using the Born approximation. Sato (1984) extended the envelope synthesis of scalar waves to the the whole envelope synthesis of 3-component seismograms from the P onset to S coda on the bases of the single scattering approximation of the RTT. His syntheses well explain how seismogram envelopes in different back azimuths vary depending on the source fault mechanism. Extension to the multiple scattering case was developed by using Stokes parameters (e.g. Margerin et al., 2000; Margerin, 2005; Przybilla et al., 2009; Sanborn et al., 2017). We also note that the Monte Carlo simulation was developed to solve stochastically the RTT (e.g. Hoshiba et al., 1991; Gusev and Abubakirov, 1987; Yoshimoto, 2000). For the data processing, it is more appropriate to stack MS envelopes of observed seismograms for comparison with the averaged intensity time traces stochastically synthesized by the RTT (e.g. Shearer and Earle, 2004; Rost et al., 2006; da Silva et al., 2018) .

When the center wavenumber of a wavelet increases much larger than the corner wavenumber of the PSDF, the wavelet around the peak value is mostly composed of narrow angle scattering around the forward direction. In such a case, the Born approximation becomes inappropriate; however, the phase shift modulation based on the parabolic approximation is useful, which is called the phase screen approximation. As an extension of the RTT with the phase screen approximation, the Markov approximation was also used for the analysis of envelope broadening and peak delay with increasing travel distance (e.g. Sato, 1989; Saito et al., 2002; Takahashi et al., 2009). Kubanza et al. (2007) measured regional differences in the lithospheric heterogeneity from the partitioning of seismic energy of tele-seismic P waves into the vertical and transverse components based on the Markov approximation.

There have been various kinds of measurements of the PSDF of the random velocity fluctuation, where the PSDF is often supposed to be a von Kármán type. In the following section, the main objective is to compile reported PSDF measurements in various scales in different geological environments of the solid earth: photo scanning of small rock samples, acoustic well logs, array analyses of tele-seismic waves; waveform analyses using FD simulations, analyses of seismogram envelopes on the basis of the RTT. We enumerate their statistical parameters and plot their PSDFs against wavenumber. We will show that the envelope of all the PSDFs is well approximated by a power-law decay curve. Then, we will discuss the results obtained and a few problems in the envelope synthesis theory for such random media and the geophysical origin of such power spectra.

## 2  Statistical characterization of random media

We consider the propagation of scalar waves as a simple model, where the inhomogeneous velocity is given by $V(\boldsymbol{x}) = V_0(1+\xi(\boldsymbol{x}))$. The fractional fluctuation $\xi(\boldsymbol{x})$ is supposed to be a random function of space. We imagine an ensemble of random media $\{\xi(\boldsymbol{x})\}$, where $\langle\xi\rangle = 0$. Angular brackets mean the ensemble average. We suppose that random media are homogeneous and isotropic, then we statistically characterize them by using the auto-correlation function (ACF):

$$R(\boldsymbol{x}) = R(r) = \langle\xi(\boldsymbol{y})\xi(\boldsymbol{y}+\boldsymbol{x})\rangle, \tag{1a}$$

where $r = |\boldsymbol{x}|$. The MS fractional fluctuation as a measure of the strength of randomness is supposed to be small, $\varepsilon^2 \equiv R(0) \ll 1$. The Fourier transform of ACF gives the PSDF:

$$P(\boldsymbol{m}) = P(m) = \iiint\limits_{-\infty}^{\infty} R(\boldsymbol{x})e^{-i\boldsymbol{m}\boldsymbol{x}}d\boldsymbol{x}, \tag{1b}$$

where wavenumber $m = |\boldsymbol{m}|$. In some literature, $(2\pi)^{-3}$ is used as a prefactor in the righthand side of (1b).

### 2.1  Several types of random media

There are several types of PSDF and ACF characterized by a few parameters.

**von Kármán-type**

The ACF is written by using a modified Bessel function of the second kind of order $\kappa$ and characteristic length $a$:

$$R(r) = \frac{2^{1-\kappa}}{\Gamma(\kappa)}\varepsilon^2 \left(\frac{r}{a}\right)^{\kappa} K_{\kappa}\left(\frac{r}{a}\right) \quad \text{for} \quad \kappa > 0, \tag{2a}$$

which is an exponential type $R(r) = \varepsilon^2 e^{-r/a}$ when $\kappa = 1/2$. In the case of space dimension $d$, the PSDF is

$$P(m) = \frac{2^d \pi^{\frac{d}{2}}\Gamma(\kappa+\frac{d}{2})\varepsilon^2 a^d}{\Gamma(\kappa)\left(1+a^2m^2\right)^{\kappa+\frac{d}{2}}} \quad \text{for} \quad \kappa > 0$$

$$\propto m^{-2\kappa-d} \quad \text{for} \quad m \gg a^{-1}. \tag{2b}$$

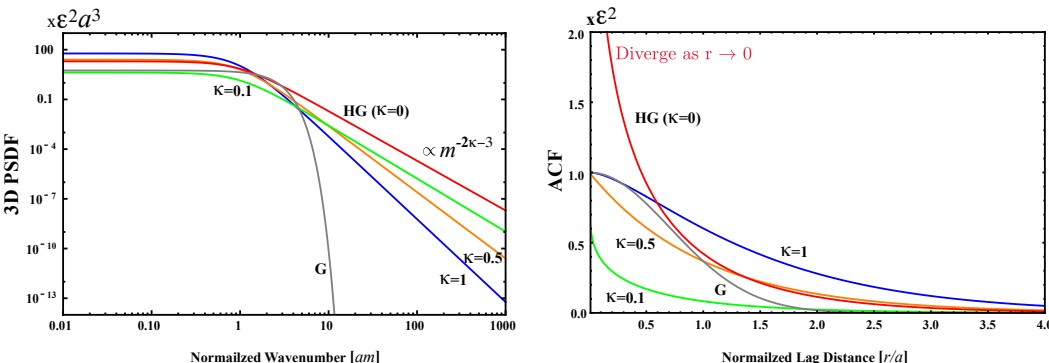

**Figure 1.** (a) Log-log plot of PSDF vs. wavenumber $m$ in 3-D space (von Kármán-type, $\kappa$=0.1, 0.5 and 1; Henyey-Greenstein type, HG, $\kappa$=0; Gaussian type, G). (b) Linear plot of ACF vs. lag distance $r$.

The PSDF obeys a power-law decay at large wavenumbers: $P(m) \propto m^{-2\kappa-3}$ for the 3D case and $P(m) \propto m^{-2\kappa-1}$ for the 1D case, where $\kappa$ corresponds to the Hurst number. In the following we will basically use a von Kármán-type for characterizing the earth medium heterogeneity.

Especially for an anisotropic case, we define the von Kármán-type PSDF in 3D (e.g. Wu et al., 1994; Nakata and Beroza, 2015):

$$P(\boldsymbol{m}) = \frac{2^3 \pi^{\frac{3}{2}} \Gamma(\kappa + \frac{3}{2}) \varepsilon^2 a_x a_y a_z}{\Gamma(\kappa) \left(1 + a_x^2 m_x^2 + a_y^2 m_y^2 + a_z^2 m_z^2\right)^{\kappa + \frac{3}{2}}} \quad \text{for} \quad \kappa > 0. \tag{3}$$

### Henyey-Greenstein type

For a case formally corresponding to $\kappa = 0$ of the von Kármán-type PSDF, we define the Henyey-Greenstein type ACF and PSDF in 3D (Henyey and Greenstein, 1941):

$$R(r) = \varepsilon^2 K_0 \left(\frac{r}{a}\right), \tag{4a}$$

$$P(m) = \frac{2\pi^2 \varepsilon^2 a^3}{(1 + a^2 m^2)^{3/2}} \approx 2\pi^2 \varepsilon^2 m^{-3} \quad \text{for} \quad m \gg a^{-1}. \tag{4b}$$

Note that parameter $\varepsilon^2$ characterizes $P$ but $\varepsilon^2 \neq R(0)$ since $R(r)$ diverges as $r \to 0$.

### Gaussian-type

Gaussian-type ACF and PSDF are also used because they are mathematically tractable.

$$R(r) = \varepsilon^2 e^{-\frac{r^2}{a^2}}, \tag{5a}$$

$$P(m) = \sqrt{\pi^3} \varepsilon^2 a^3 e^{-\frac{m^2 a^2}{4}}. \tag{5b}$$

We plot those PSDFs against wavenumber and ACFs against lag distance in Figure 1.

## 3 Measurements of random heterogeneities

There are several kinds of measurements for estimating statistical parameters characterizing random media. Here we principally collect measurements supposing a von Kármán type for isotropic randomness; however, we include a few measurements supposing anisotropic randomness and a Gaussian type. In a small scale, the photo scan method is applied to small rock samples. Acoustic well logs are available in deep wells drilled in the shallow crust. When the precise velocity tomography result is available, we can directly calculate the PSDF. In seismology, the most conventional method is to analyze seismograms of natural earthquakes or artificial explosions after traveling through the earth heterogeneity. It is better to focus on MS amplitude envelopes (intensity time traces) since phases are complex caused by random heterogeneities. Comparing observed seismogram envelopes with envelopes synthesized in random media, we can evaluate von Kármán parameters. For the synthesis, we can use the finite difference simulation (FD), the RTT with the Born approximation, and the RTT with the phase screen approximation that is equivalent to the Markov approximation. For each reported measurement, we enumerate the target region, data and the method, the measured PSDF as a function of wavenumber $m$, von Kármán parameters ($\kappa$, $\varepsilon$, $a$), the frequency range, the wavenumber range, and the reference in Tables 1∼3. Note that measurements of heterogeneity listed in the Tables are by no means the only ones. Especially in seismological measurements, we estimate the wave number range from the frequency range by using the typical velocity of the target medium. In the Tables, the parameter value in brackets ($\cdots$) is a priori fixed in the measurement. Then, we plot obtained PSDFs against wavenumber in Figures 2∼5. When the estimated parameter value is given by a range, a value in squared brackets [$\cdots$] is used as a representative for plotting PSDFs in the Figures. Measurement of a label with an asterisk * is insufficient for plotting the PSDF in the Figures.

### 3.1 Photo scan of the rock surface

The photo scan method uses a scanner to take a picture of the polished flat surface of a small rock sample (e.g. Sivaji et al., 2002; Spetzler et al., 2002; Fukushima et al., 2003). For the case of a granite sample, they classified color images on a straight line into three types of mineral grains; quartz, plagioclase and biotite. Assigning a typical velocity $V_P$ or $V_S$ to each mineral grain, they constructed a velocity profile along the line. Then, they estimated the 1D PSDF of the velocity fractional fluctuation. They measured 1D PSDFs of granite and gabbro samples fixing $\kappa$=0.5 as R1∼R5. Figure 2 (a) shows estimated 1D PSDFs, where the wavenumber range is of the order of $1\,\mathrm{mm}^{-1}$. We note that raw 1D PSDFs in Figures 4 and 5 of Fukushima et al. (2003) decay a little slower than those of R4 and R5 in Figure 2 (a) especially at large wavenumbers.

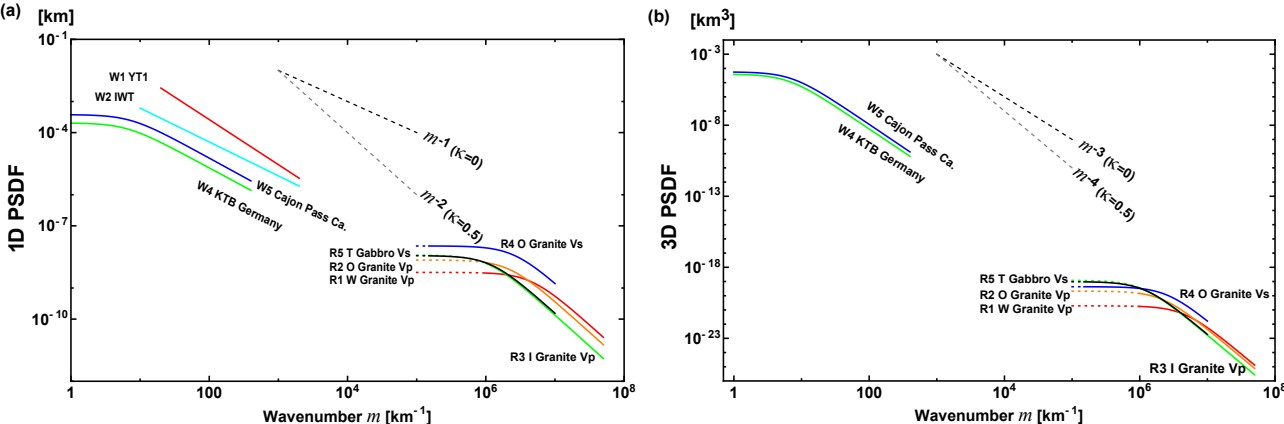

**Figure 2.** (a) 1D PSDF vs. wavenumber for rock samples and acoustic well logs. (b) Converted 3D PSDF vs. wavenumber, where the randomness is supposed to be isotropic. See labels in Table 1.

### 3.1.1 Conversion from 1D PSDF into 3D PSDF

In the case of isotropic randomness, we evaluate the 1D PSDF from the 3D ACF along the $z$-axis at $x = y = 0$ as follows:

$$P_{1D}(m_z) \equiv \int_{-\infty}^{\infty} R_{3D}(0,0,z)\, e^{-im_z z}\, dz = \int_{-\infty}^{\infty} \left[ \frac{1}{(2\pi)^3} \iiint_{-\infty}^{\infty} P_{3D}(m_x', m_y', m_z')\, e^{im_z' z}\, d\mathbf{m}' \right] e^{-im_z z}\, dz$$

$$= \frac{1}{(2\pi)^2} \iint_{-\infty}^{\infty} P_{3D}(m_x', m_y', m_z)\, dm_x' dm_y'. \tag{6a}$$

Substituting (2b) into the above equation, we have

$$P_{1D}(m_z) = \frac{1}{(2\pi)^2} \iint_{-\infty}^{\infty} \frac{8\pi^{3/2}\varepsilon^2 a^3 \Gamma(\kappa + 3/2)}{\Gamma(\kappa)\left[1 + a^2\left(m_x'^2 + m_y'^2 + m_z^2\right)\right]^{\kappa+3/2}}\, dm_x' dm_y' = \frac{2\pi^{1/2}\Gamma(\kappa+1/2)\varepsilon^2 a}{\Gamma(\kappa)\left(1 + a^2 m_z^2\right)^{\kappa+1/2}}. \tag{6b}$$

Thus, we can evaluate the 3D PSDF from the 1D PSDF using parameters $\varepsilon$, $\kappa$ and $a$ of 1D PSDF.

Supposing the randomness is isotropic, we evaluate corresponding 3D PSDFs of R1~R5 and plot them in Figure 2 (b).

### 3.2 Acoustic well loggings in boreholes

An acoustic well log is obtained from the measurement of the travel time of an ultrasonic pulse along the wall of a borehole. Measurements W1 (volcanic tuff) and W2 (tertiary to pre-tertiary) in Japan clearly show power law decay with $\kappa = 0.225$ and 0.045, respectively; however, a corner is not clearly seen in each PSDF. Measurement W4 at the deep well KTB in Germany shows $\kappa = 0.10$. Measurement W3 in the same well shows that the exponent of wavenumber is –0.97, which formally corresponds to a negative $\kappa$. Measurement W5 at Cajon pass in California shows $\kappa = 0.11$. All these measurements show very

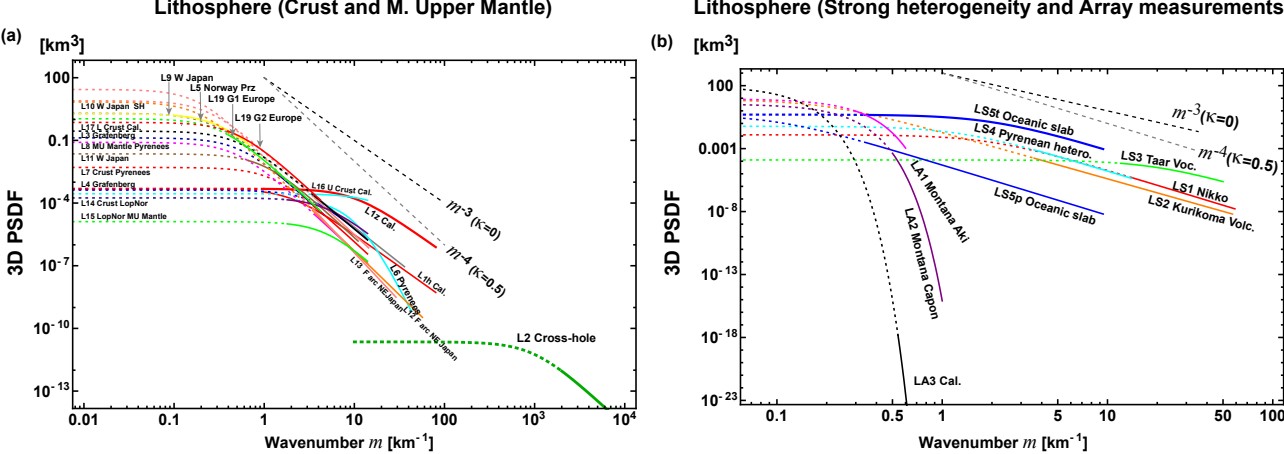

**Figure 3.** 3D PSDF vs. wavenumber for (a) the lithosphere (the crust and most upper mantle), (b) strong heterogeneities and array data analyses in the lithosphere. See labels in Table 2.

small $\kappa$ values close to 0. Shiomi et al. (1997) made a list of reported exponents of wavenumber, which shows that most of $\kappa$ values are smaller than 0.25. Measurement of $a$ seems to be restricted by the sample length. We enumerate those measurements in Table 1 and plot their 1D PSDFs against wavenumber in Figure 2 (a). Figure 2 (b) plots the corresponding 3D PSDFs of W4 and W5.

We note that Wu et al. (1994) measured anisotropy of randomness from the analysis of well-logs obtained from two parallel wells at KTB: the ratio of characteristic scales in horizontal to vertical directions $a_h/a_z$=1.8 (see (3)) as shown in W3.

### 3.3   Velocity tomography

There have been measurements of velocity tomography in various scales, from which we can calculate the PSDF and then estimate von Kármán-type parameters. This method depends on the spatial resolution of tomography result. Measurement L1

in Table 2 is calculated from the precise $V_P$ tomography result of the shallow crust, Los Angeles, California: the exponent of wavenumber is –3.08 ($\kappa = 0.04$). Anisotropic randomness is also reported: $a_z$=0.1 km and $a_h$=0.5 km (see eq. (3)). We show those in Figures 3 (a). Measurement M2 in Table 3 is evaluated from the 2D PSDF of $V_S$ tomography result of the upper mantle in a low wavenumber range. Although there is a resolution limit of the tomography method, the exponent of wavenumber is between –2 and –3, which means $0 < \kappa < 0.5$. We note that Figure 8 of Mancinelli et al. (2016a) shows that the 1D PSDF

estimated from the $V_P$ tomography result in the upper mantle (Meschede and Romanowicz, 2015) well covers that of MU2 ($\kappa$=0.05, $\varepsilon$=0.1, $a$=2000 km) for the wavenumber range $2 \times 10^{-4} \sim 10^{-2}$ km$^{-1}$.

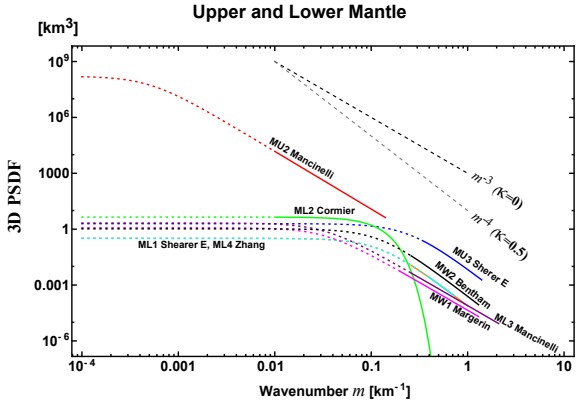

**Figure 4.** 3D PSDF vs. wavenumber for the upper and lower mantle. See labels in Table 3.

## 3.4   Array analysis of tele-seismic P waves

Tele-seismic P waves registered by a large aperture array were used for the evaluation of the 3D PSDF of the lithosphere beneath the array: LA1 and LA2 of Table 2 in Montana and LA3 in southern California used amplitude and phase coherence analyses, where a Gaussian-type PSDF (eq. (5)) was supposed because of mathematical simplicity. As shown in Figure 3 (b), they drop very fast as wavenumber increases. Later Flatté and Wu (1988) developed the angular coherence analysis in addition to the above methods. Analyzing tele-seismic P waves registered at NORSAR, they proposed a two overlapping layer model LA4, which is composed of a band-limited flat spectrum from the surface to 200 km in depth and $m^{-4}$ spectrum ($\kappa = 0.5$, $\varepsilon = 1 \sim 4\%$) for depths from 15 to 250 km. It means $\kappa < 0.5$ and the roll-off of their PSDF is much smaller than that of Gaussian types (not shown in Figure 3 (b)).

## 3.5   Finite difference simulations

The finite difference (FD) simulation is often used for the numerical simulation of waves in an inhomogeneous velocity structure. For the evaluation of average MS amplitude envelopes, we have to repeat simulations of the wave propagation through random media having the same PSDF that are generated by using different random seeds. There are several measurements of statistical parameters using FD as L9∼L11 and ML4 in Tables 2 and 3. Measurement LS5 focused on the fact that the subducting oceanic plate is an efficient waveguide for high-frequency seismic waves: estimated anisotropic parameters are $\kappa$=0.5, $\varepsilon$=0.02, $a_p$=10 km and $a_t$=0.5 km in the parallel and transverse directions, respectively. Note ML2 supposes a Gaussian-type.

## 3.6   Analyses of seismogram intensities (MS amplitude envelopes)

The RTT is essentially stochastic to synthesize directly the intensity (the average MS amplitude envelope) of a wavelet propagating through random media. There are two conventional methods on the basis of the RTT: one uses the Born approximation

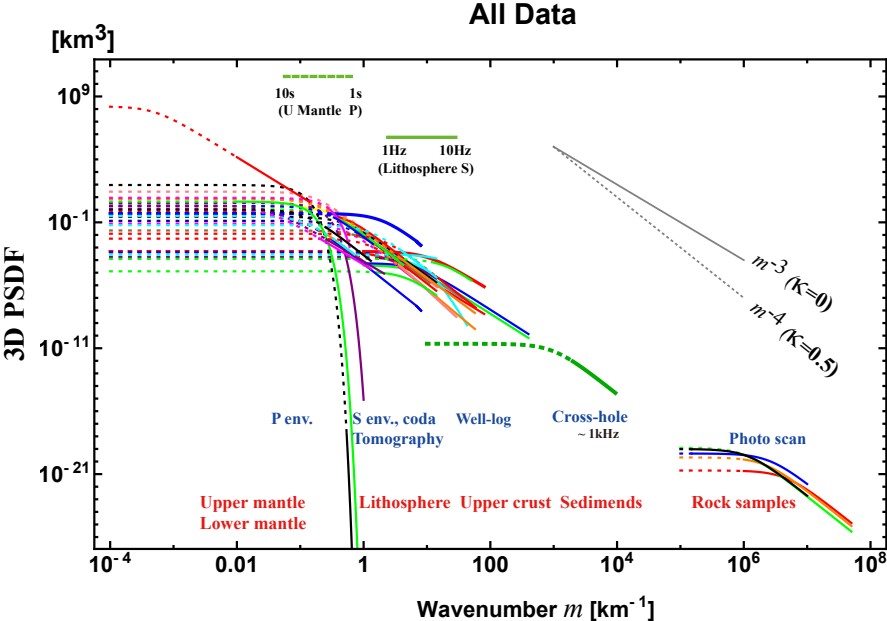

**Figure 5.** 3D PSDF vs. wavenumber for all the data.

and the other uses the phase screen approximation based on the parabolic approximation when the wavenumber is larger than the corner. The former neglects the phase shift, but the latter correctly considers the phase shift.

### 3.6.1 Scalar wave scattering by a single obstacle

We here study the deterministic scattering of scalar waves by a single spherical obstacle (radius $a = 5$ km and velocity anomaly
$\varepsilon = +0.05$) embedded in a homogeneous medium ($V_0$ =4 km/s) as a mathematical model. The Born approximation calculates spherically outgoing scattered waves putting the incident plane wave of wavenumber $k_c$ in the interaction term of the first order perturbation equation. From the scattering amplitude we evaluate the total scattering cross-section $\sigma_0$ as a measure of scattering power of the obstacle. The resultant $\sigma_0$ monotonously increases with frequency as shown by a blue line in Figure 6. As the wavenumber increases ($ak_c \gg 1$), the phase shift increases as the incidence plane wave penetrates the obstacle. Putting
the phase modulated wave according to the parabolic approximation (the phase screen approximation) into the interaction term of the first order perturbation equation, we calculate the scattering amplitude and then the total scattering cross-section. It is the distorted-wave Born approximation with the phase screen approximation, which is also referred to as the Eikonal approximation. This approximation predicts that $\sigma_0$ (a red line in Figure 6) saturates at high frequencies and converges to $2\pi a^2$, which is twice the geometrical section area of the obstacle as predicted by shadow scattering (e.g. Landau and Lifshitz, 2003,
p. 519 and 543). We recognize that the conventional Born approximation is still accurate even for $ak_c > 1$; however, it works well only for $\varepsilon^2 a^2 k_c^2 \lesssim O(0.1)$. We should use the distorted-wave Born approximation with the phase screen approximation

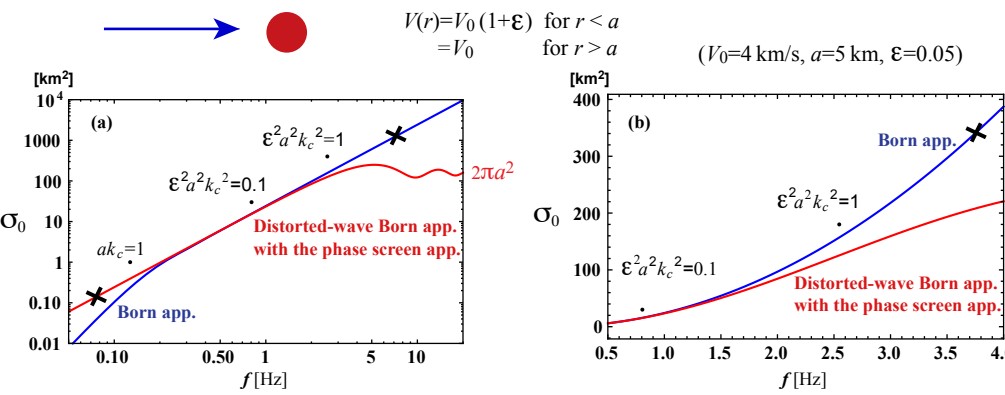

**Figure 6.** Deterministic scattering of scalar waves by a high velocity sphere. (a) Log-log plot of the total scattering cross-section against frequency. (b) Semi log plot for zoom up. The Born approximation and the distorted-wave Born approximation with the phase screen approximation are drawn by blue and red lines, respectively.

for $\varepsilon^2 a^2 k_c^2 \gtrsim O(1)$. The two approximations predict nearly the same $\sigma_0$ value in the intermediate range. We note that $2\varepsilon a k_c$ is the phase shift on the center line after passing the obstacle. Note that the phase screen approximation is not applicable for $a k_c < 1$ since it is based on the parabolic approximation.

Interpreting $\varepsilon$ and $a$ as the RMS fractional fluctuation and the characteristic length of uniformly distributed random media, we may use the inequality $\varepsilon^2 a^2 k_c^2 \ll O(1)$ or $\varepsilon^2 a^2 k_c^2 \lesssim O(0.1)$ as a criterion of the Born approximation used in the RTT.

### 3.6.2 RTT with the Born approximation

For uniformly distributed random media characterized by $P(m)$, the Born approximation leads to the scattering coefficient at wavenumber $k_c$ into scattering angle $\psi$:

$$g(k_c, \psi) = \frac{k_c^4}{\pi} P(2k_c \sin \frac{\psi}{2}), \tag{7a}$$

which is axially symmetric. The total scattering coefficient is

$$g_0(k_c) \equiv \frac{1}{4\pi} \oint g(k_c, \psi) d\Omega = \frac{1}{2} \int_0^\pi g(k_c, \psi) \sin \psi d\psi = \int_0^{2k_c} g_{ker}(k_c, m) dm, \tag{7b}$$

where $m = 2k_c \sin \frac{\psi}{2}$. The integral kernel in the wavenumber space is given by

$$g_{ker}(k_c, m) = \frac{k_c^2}{2\pi} m P(m). \tag{7c}$$

The upper bound of the integral is twice the wavenumber. As an example, Figure 7 shows plots of $P(m)$ (blue) vs. $m$ and $g_{ker}(m)$ vs. $m$ at 0.1 Hz (red) and 1 Hz (green) for the case of $\kappa$=0.5, $\varepsilon$=0.05, $a$=1 km and $V_0$=4 km/s. As shown at the

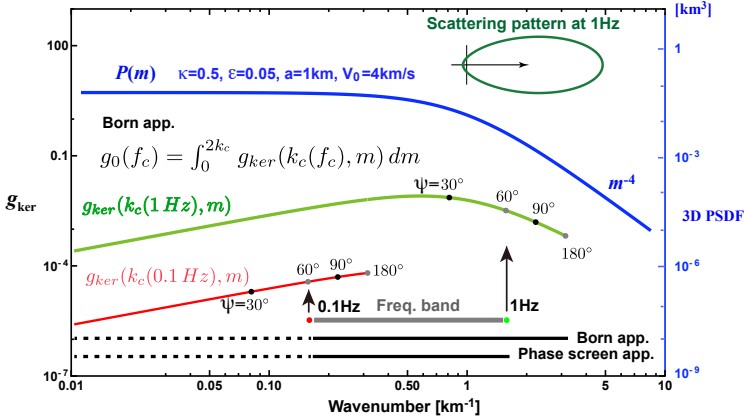

**Figure 7.** Plot of $P(m)$ (blue, right scale) and the spectral kernel of the scattering coefficient $g_{ker}(k_c(f_c), m)$ at $f_c$=0.1 Hz (red, left scale) and 1 Hz (green, left scale) according to the Born approximation. Scattering angles are marked by dots on each trace. For the case of frequency band between 0.1 and 1 Hz, the phase screen approximation based on the parabolic approximation covers the wavenumber range from 0 to the upper bound (line at the bottom), however, the Born approximation covers the range from 0 to twice the upper bound (line next to the bottom). We use those line styles in Figures 3∼5 and 9.

upper-right corner, the scattering pattern at 1 Hz has a large lobe into the forward direction; however, it becomes isotropic as the frequency decreases. Dots on each $g_{ker}$ curve show corresponding scattering angles.

In the framework of the RTT, the Monte Carlo simulation is a simple method to synthesize stochastically the wavelet intensity time trace. A particle carrying unit intensity is shot randomly from a point source, and its trajectory is traced with the increment of time steps. The occurrence of scattering is stochastically tested by inequality $g_0 V_0 \Delta t >$ Random[0,1] at every time step of $\Delta t$, and $g(k_c, \psi)/(4\pi g_0(k_c))$ is used as the probability of scattering into angle $\psi$. Note that Random[0,1] is a uniform random variable between 0 and 1. Since $g_0 V_0 \Delta t$ is chosen to be small enough, scattering does not occur every time step but intermittently. As a simple example, Figure 8 (a) schematically illustrates the flowchart of the Monte Carlo simulation for the isotropic radiation from a point source in uniform random media. At lapse time $t$, dividing the number of particles $n$ registered in a spherical shell of radius $r$ and a thickness $\Delta r$ by the total number of particles $N$ and the shell volume $4\pi r^2 \Delta r$, we calculate the intensity Green function $G(r, t)$. The intensity time trace $I(r, t)$ is calculated by the convolution of $G(r, t)$ and the source intensity time function $S(t)$ in the time domain. It is easy to introduce a layered structure of background velocity and intrinsic absorption into the simulation code.

The RTT for the scalar wave case can be extended to the elastic vector wave case by using Stokes parameters. There are four scattering modes, PP, PS, SS and SP scatterings, and the S-wave scattering coefficients are not axially symmetric (see Sato et al., 2012, Figure 4.7). Many papers (e.g. Shearer and Earle, 2004; Przybilla et al., 2009) suppose proportional relations

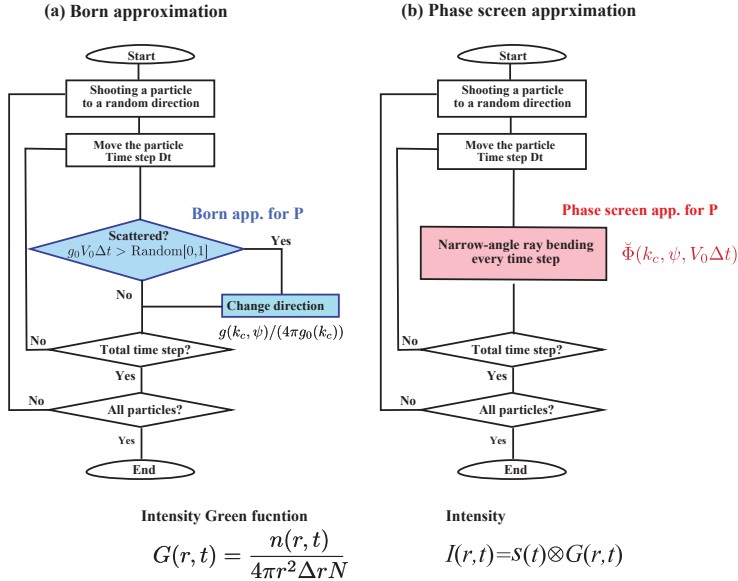

**Monte Carlo simulation diagam according to the RTT**

**(a) Born approximation**

Start

Shooting a particle to a random direction

Move the particle Time step Dt

**Born app. for P**

Scattered? $g_0 V_0 \Delta t > \text{Random}[0,1]$ — Yes

No

Change direction $g(k_c, \psi)/(4\pi g_0(k_c))$

Total time step? — No / Yes

All particles? — No / Yes

End

Intensity Green fucntion

$$G(r,t) = \frac{n(r,t)}{4\pi r^2 \Delta r N}$$

**(b) Phase screen apprximation**

Start

Shooting a particle to a random direction

Move the particle Time step Dt

**Phase screen app. for P**

Narrow-angle ray bending every time step — $\breve{\Phi}(k_c, \psi, V_0 \Delta t)$

Total time step? — No / Yes

All particles? — No / Yes

End

Intensity

$$I(r,t) = s(t) \otimes G(r,t)$$

**Figure 8.** Flowchart of the Monte Carlo simulation code according to the RTT for the scalar wavelet intensity in uniform random media. (a) RTT with the Born approximation. (b) RTT with the phase screen approximation.

$\delta V_p/V_{P0} = \delta V_S/V_{S0} = \xi$ and $\delta\rho/\rho_0 = \nu\xi$ based on the empirical Birch's law, which reduce three fractional fluctuations into one (e.g. Sato et al., 2012, eqs. 4.58 and 4.59).

The RTT with the Born approximation has been often used not only for the analyses of S coda envelopes but also for the whole seismogram envelope from the P onset via P coda through S wave until S coda (see Tables 2 and 3). This method has

5 been often used not only for the analyses of regional seismograms propagating through the lithosphere but also for the analyses of tele-seismic waves propagating through the mantle. This method is not only applied to direct P phase but also PcP and $\text{PKP}_{prec}$ phases and so on. In this review, we neglect intrinsic attenuation parameters a priori assumed or measured in each paper. For a given wavenumber range $(k_l, k_u)$ (gray) in Figure 7, each PSDF curve using this method in Figures 3∼5 and 9 is drawn by a dotted line for $(0, k_l)$ and a solid line for $(k_l, 2k_u)$ as the line next to the bottom of Figure 7. As indicated by dots

on the $g_{ker}$ curves, the wavenumber interval of solid line reflects wide angle scattering and that of dotted line reflects narrow angle scattering around the forward direction.

Most of measurements of S-waves in the lithosphere cover the wavenumber range up to $100\,\text{km}^{-1}$. Measurement L2 analyzed cross-hole seismograms of the order of kHz by using 2D-RTT, of which the wavenumber range is as high as the order of $1\,\text{m}^{-1}$. Measurement MU2 for tele-seismic P wave envelopes at long periods in the upper mantle shows that the characteristic

scale $a$=2000 km is much larger than those of MU3 and ML1 at short periods. Several measurements a priori suppose $\kappa$ =0.5; however, most of measurements show $\kappa < 0.5$ except L6. Measurements ML3 and MW1 propose the H-G type (see eq. (4))

corresponding to $\kappa = 0$ for the lower/whole mantle. We note that Mancinelli et al. (2016b) proposed an alternative model of 3D-PSDF $\propto m^{-2.6}$ in addition to ML3 (not shown in Figure 4 ).

### 3.6.3 RTT with the phase screen approximation

When $ak_c \gg 1$, scattering mostly occurs within a narrow angle around the forward direction. At a large travel distance, the wavelet just after the onset is mostly composed of those waves. The phase screen approximation correctly calculates the phase shift modulation. For the incidence of a plane wave into the $z$ direction, the mutual coherence function (MCF) of the phase shift modulated waves for an increment $\Delta z$ is given by

$$\Phi(k_c, r_\perp, \Delta z) = e^{-k_c^2(A(0) - A(r_\perp))\Delta z}. \tag{8a}$$

The longitudinal integral of the ACF is

$$A(r_\perp) = \int_{-\infty}^{\infty} R(\boldsymbol{x}_\perp, z)dz = \frac{1}{(2\pi)^2} \iint_{-\infty}^{\infty} P(\boldsymbol{m}_\perp, m_z = 0)e^{i\boldsymbol{m}_\perp \boldsymbol{x}_\perp} d\boldsymbol{m}_\perp, \tag{8b}$$

where $\boldsymbol{x}_\perp$ is the transverse coordinate vector and $r_\perp = |\boldsymbol{x}_\perp|$ (Sato et al., 2012, eq. 9.60). Taking the Fourier transform of MCF $\Phi$ with respect to transverse coordinates, we have

$$\breve{\Phi}(k_c, k_\perp, \Delta z) = \frac{1}{(2\pi)^2} \iint_{-\infty}^{\infty} \Phi(k_c, r_\perp, \Delta z)e^{i\boldsymbol{k}_\perp \boldsymbol{x}_\perp} d\boldsymbol{x}_\perp \xrightarrow[\Delta z \to 0]{} \delta(\boldsymbol{k}_\perp). \tag{8c}$$

Since $\iint_{-\infty}^{\infty} \breve{\Phi}(k_c, k_\perp, \Delta z)d\boldsymbol{k}_\perp = 1$, interpreting $\breve{\Phi}(k_c, k_\perp, \Delta z)$ as the probability of ray bending angle $\psi = \tan^{-1} \frac{k_\perp}{k_c}$ per increment $\Delta z = V_0\Delta t$, we can stochastically synthesize the intensity by using the Monte Carlo simulation (e.g Williamson, 1972; Takahashi et al., 2008; Saito et al., 2008). As a simple example, Figure 8 (b) schematically illustrates the flowchart of the RTT with the phase screen approximation for the isotropic radiation from a point source in uniform random media. Different from the Born approximation, narrow-angle ray bending occurs at every time step. The intensity Green function can be obtained in the same manner as the RTT with the Born approximation. This approximation well synthesizes the intensity time trace having a delayed peak from the onset and a decaying tail of early coda at large travel distances. This approximation can not synthesize the late coda intensity since wide angle scattering is neglected. The Markov approximation is known as a stochastic extension of the phase screen method for the two-frequency mutual coherence function (e.g. Saito et al., 2002). If we focus on the intensity time trace around the peak arrival, the Markov approximation and the RTT with the phase screen approximation show good coincidence (see Sato and Emoto, 2018, Figure 8).

When this approximation is used, $k_c \gg a^{-1}$ is a priori supposed. Most of this type of measurements read the peak delay and the envelope width of each seismogram envelope. There is a merit that the peak delay measurement is rather insensitive to intrinsic absorption. In the NE Japan, $\kappa$ value beneath a volcano LS2 is smaller than those in the fore-arc side L12 and L13. Note that narrow angle scattering around the forward direction dominates in tele-seismic wavelets even if the Born approximation is used for the analysis. Narrow angle scattering is mostly produced by the PSDF in low wavenumbers compared with $k_c$. For a

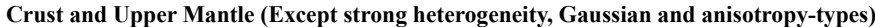

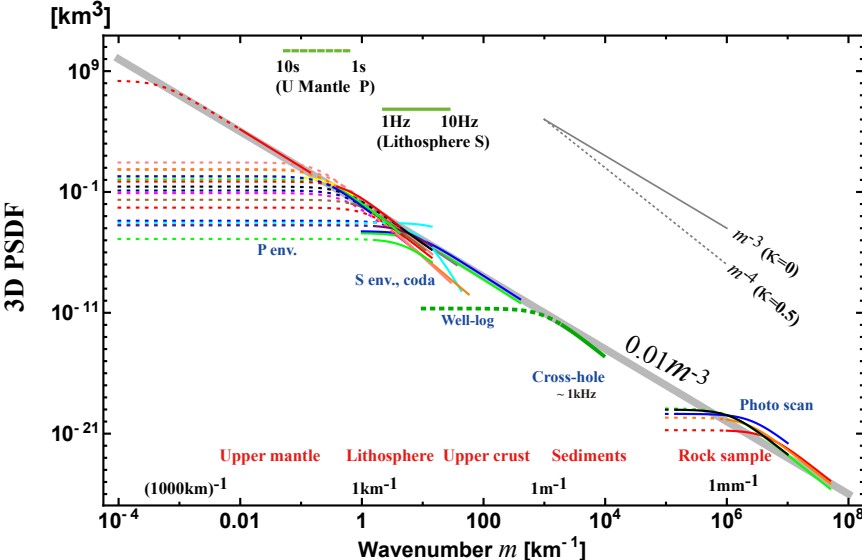

**Figure 9.** 3D PSDF vs. wavenumber for the crust and the upper mantle. Data of Gaussian-type, anisotropy type, strong heterogeneity, the lower mantle, and the whole mantle are excluded. The light gray straight line visually well fits to most of spectral envelopes.

given wavenumber range $(k_l, k_u)$ (gray) in Figure 7, each PSDF curve using this method in Figure 3 is drawn by a dotted line for $(0, k_l)$ and a solid line for $(k_l, k_u)$ as the bottom line of Figure 7.

### 3.7 Characteristics of reported PSDFs

#### 3.7.1 All the data

Some measurements a priori assumed $\kappa = 0.5$; however, most of measurements report $\kappa < 0.5$. In the mantle, $\kappa$ is very small close to zero and an H-G type is also proposed. The RMS fractional fluctuation $\varepsilon$ is of the order of 0.1 for rock samples and well log data and of the order from 0.01 to 0.1 in the lithosphere and the upper mantle. Large values are reported for the shallow crust L16 and beneath a volcano LS3, however, smaller values are reported for the lower mantle. The characteristic scale $a$ varies a lot depending on measurements. The corner wavenumber $a^{-1}$ is not clearly seen in PSDFs of acoustic well logs. Some measurements report anisotropy: W3 of well-logs, L1 of velocity tomography in the shallow crust and LS5 in the subducting oceanic slab. The characteristic length in the vertical direction is smaller than the horizontal direction in the shallow crust, and that in the transverse direction is smaller than that in the direction parallel to the subducting slab.

   Plotting PSDFs against wavenumber is more informative for understanding the random heterogeneity compared with enumerating statistical parameter values. Figure 5 shows the plot of 3D PSDF vs. wavenumber for all the data covering a wide wavenumber range $10^{-4} \sim 10^8\,\text{km}^{-1}$. We recognize that Gaussian type PSDFs show very different behavior from others,

which suggests Gaussian type assumption is inappropriate. PSDFs in the lower mantle take smaller values, and those for volcanoes and for the shallow crust take larger values than others.

### 3.7.2 Lithosphere and the upper mantle except strong heterogeneity, Gaussian and anisotropy-types

Eliminating data supposing a Gaussian type LA1∼LA3, strong heterogeneity data LS1∼LS4, anisotropy type data L1 and LS5, and the lower mantle and the whole mantle data ML1∼ML4 and MW1∼MW2 from Figure 5, we plot the rest of data for the crust and the upper mantle in Figure 9. Most of $\varepsilon$ values are of the order of $0.01 \sim 0.1$, most of $\kappa$ values are less than or equal to 0.5 and many of them are close to 0, and the high wavenumber end of the power-law decay branch of each PSDF is not far from each corner wavenumber.

We draw a power-law decay line $\mathrm{PSDF}(m) = 0.01 m^{-3} \,\mathrm{km}^3$ (gray) visually fitting to most of PSDF envelopes for a very wide range of wavenumbers $10^{-3} \sim 10^8 \,\mathrm{km}^{-1}$. This line is not the average of PSDFs. This line looks like an extension of MU2 in the upper mantle into higher wavenumbers of the shallow crust.

## 4 Discussions

### 4.1 Measurements

It will be necessary for us to measure the small-scale randomness of sedimentary rock samples. More measurements are necessary in the wavenumber range $10^3 \sim 10^5 \,\mathrm{km}^{-1}$ since there are few measurements.

In most of PSDF measurements, each power-law decay branch is short since the Born approximation senses the spectrum up to twice the wavenumber. It will be necessary to measure how each power-law decay branch varies with wavenumber increasing. It will be necessary to estimate the corner $a^{-1}$ in each measurement with a wide wavenumber-range covering sufficiently large enough the both sides of the corner. The flat part, the low-wavenumber side of each PSDF drawn by a dotted line in Figures is also important as the cause of narrow angle scattering.

Although most of measurements used in this review analyzed intrinsic attenuation; however, we did not enumerate them in this review since different assumptions were used in different measurements. It will be necessary for us to measure systematically the PSDF of random heterogeneity in conjunction with intrinsic attenuation.

We should note that there are large variations in $\delta \ln V_S / \delta \ln V_P$ and $\nu \equiv \delta \ln \rho / \delta \ln V_S$ in the earth. Koper et al. (1999) estimated $\delta \ln V_S / \delta \ln V_P$ to be in the range 1.1∼1.5 in the Tonga Slab. Romanowicz (2001) estimated $\delta \ln V_S / \delta \ln V_P$ to be larger than 2.5 in the lower mantle at larger scale lengths. Many measurements reported use $\nu = 0.8$ for the synthesis, which is appropriate for the shallow lithosphere. Parameter $\nu$ takes smaller values as 0.17 for volcanic-tuff (Shiomi et al., 1997) and 0.31∼0.33 for sandstone and shale (Kenter et al., 2007). In the mantle, Karato (2008) estimated $\nu = 0.23 \sim 0.42$ for the S-wave velocity predicted from the temperature derivatives of seismic wave velocities and thermal expansion, and $\nu = 0.15 \sim 0.23$ including the influence of anelasticity. It will be necessary to introduce realistic $\delta \ln V_S / \delta \ln V_P$ and $\delta \ln \rho / \delta \ln V_P$ in the synthesis.

Figure 9 summarizes reported PSDF measurements supposing isotropic randomness; however, there are measurements clearly showing anisotropic randomness such as W3 and L1 for the shallow crust and LS5 for the oceanic slab. Those may reflect the effect of gravity for the creation of anisotropy. It will be necessary for us to study how a wavelet propagates through anisotropic random heterogeneity of the earth medium (e.g. Margerin, 2006).

## 4.2  Mathematical Theory

In section 3.6, we mentioned that the conventional Born approximation is inapplicable and the phase screen approximation is useful when the phase shift becomes large as the wavenumber increases. In order to avoid the difficulty, taking the center wavenumber of a wavelet as a reference, Sato and Emoto (2018) proposed to divided the PSDF into two components (see also Sato, 2016; Sato and Emoto, 2017). They use the Born and phase-screen approximations to the short-scale (high-wavenumber) and long-scale (low-wavenumber) components, $P_S$ and $P_L$, respectively, in the RTT in order to explain simultaneously the envelope broadening just after the onset and the excitation of late coda waves. Figure 10 illustrates the flowchart of their Monte Carlo simulation. Their spectrum division method looks like an implementation of the distorted-wave Born approximation in the RTT since it describes wide angle scattering for the incidence of the phase-shift modulated wave. They successfully synthesized intensity time traces that well explain FD simulation results for the case of $ak_c = 23.6$ and $\varepsilon^2 a^2 k_c^2 = 1.39$. It would be interesting to see how this method may be extended to polarized elastic waves.

We note that some papers numerically show that the RTT with the Born approximation works well in some cases over the above limitation. Przybilla et al. (2006) synthesizes vector-wave intensity that well fits to that of the FD simulation in 2D even for S-waves of $ak_c$=58 and $\varepsilon^2 a^2 k_c^2 = 8.4$ (see their Table 1) if the wandering effect is convolved as a result of the travel time fluctuation. Emoto and Sato (2018) show that the synthesized scalar intensity by the RTT with the Born approximation well fits to that of the FD simulation in 3D even for the case of $ak_c = 23.6$ and $\varepsilon^2 a^2 k_c^2 = 1.39$ when the wandering effect is convolved. If the earth heterogeneity is represented by a power-law decay power spectrum for such a wide wavenumber range, which means that the corner wavenumber is very low, we should carefully examine the applicability of the Born approximation in the RTT.

Acoustic well-logging and photo scan faithfully measure the inhomogeneous elastic coefficients. The RTT used here supposes the scattering contribution of random inhomogeneity of elastic coefficients only; however, observed seismograms do not only reflect those but also the scattering contribution of pores and cracks distributed over the earth medium. It will be necessary for us to study their contribution in the intensity synthesis.

## 4.3  Power-law decay spectral envelope

In observation, we may take the power-law spectral envelope as a reference curve for studying the regional differences especially in the power-law decay part of the PSDF. The characteristic length $a$ seems to increase as the wavenumber of a wavelet decreases or as the dimension of measurement system becomes large. It reminds us that the characteristic scale of the slip distribution increases with increasing source dimension as Mai and Beroza (2002) analyzed finite-fault source inversion results (see their Figure 12).

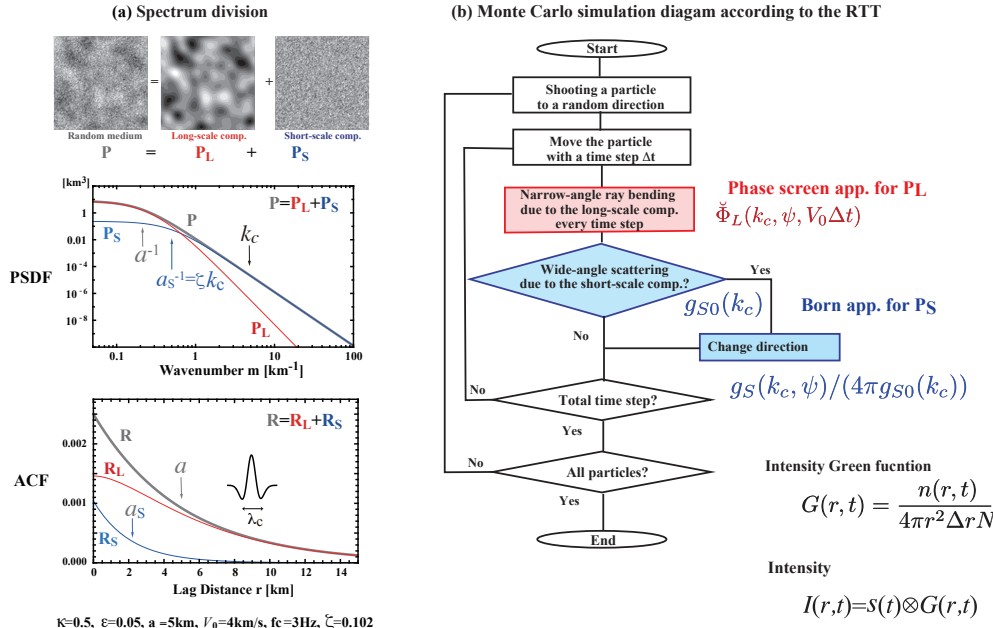

**Figure 10.** Spectrum division method. (a) Division of $P$ into two components, $P_S$ and $P_L$, with respect to the center wavenumber of a wavelet $k_c$ as a reference, where $\zeta$ is a tuning parameter. (b) Flow-chart of the Monte Carlo simulation according to to the RTT with the spectrum division method. Modified from Sato and Emoto (2018).

The power-law decay spectral envelope reminds us of the observed fractal nature of various kinds of surface topographies: Sayles and Thomas (1978a, b) show 1D-PSDF $\propto m^{-2}$ for wavelengths $10^{-6} \sim 10^3$ km although the power exponent varies from –1.07 to –3.03 for small segments; Brown and Scholz (1985) show 1D-PSDF $\propto m^{-1.64 \sim -3.36}$ for the wavenumber range $10^{-6} \sim 0.1 \ \mu m^{-1}$ especially for the topography of natural rock surfaces and faults. We also note that the PSDF of the refractive index fluctuation of air obeys the Kolmogorov spectrum: 3D-PSDF $\propto m^{-11/3}$, where $\kappa = 1/3$. This spectrum is physically produced by the cascade in the turbulent flow of low viscosity air: the large eddies breaks up originating smaller eddies dissipating energy by viscosity. However, it may be difficult to apply this cascade model to the mantle since the viscosity of mantle fluid is thought to be high.

For igneous rocks such as granite, there are variations in composition of minerals and grain sizes, which depend on a variety of slow crystallization differentiations of basaltic magma. Random variations of acoustic well-log profiles reflect the complex sedimentation process during the geological history. Volcanism produces more heterogeneous structures composed of pyroclastic material and lava. For random heterogeneities in the mantle, we imagine complex mantle flow. Mancinelli et al. (2016a) referred to a marble cake mantle model (Allègre and Turcotte, 1986) containing heterogeneity mostly composed of basalt and harzburgite in many scales in the upper mantle in order to explain the power-law spectrum. Stixrude and Lithgow-Bertelloni (2007) proposed the velocity variation due to chemical and phase stability at different depths, which is a possible candidate

especially for the heterogeneity in the vertical direction. If we accept the power-law decay spectrum, we will have to study what kinds of geophysical mechanisms created such random medium spectra in different scales and in different geological environments in the solid earth.

### 4.4 Isotropic scattering coeffcient

In advance to the measurements based on the RTT for anisotropic scattering presented here, there have been many measurements of the isotropic scattering coefficient $g_{iso}$ in the world on the basis of the RTT with the isotropic scattering assumption (e.g. Sato et al., 2012; Yoshimoto and Jin, 2008). The isotropic scattering model is mathematically tractable, and the multiple lapse-time window analysis (Fehler et al., 1992; Hoshiba, 1993) has been often used for practical analyses. This method essentially estimates $g_{iso}$ from the ratio of late coda excitation to the radiated energy irrespective of the envelope broadening. Recent measurements show that $g_{iso}$ decreases with depth (e.g. Rachman and Chung, 2016; Badi et al., 2009). It will be interesting to plot the frequency dependence of reported $g_{iso}$ for a wide range of frequencies and to study the relation with the obtained power spectral envelope shown in Figure 9.

## 5 Conclusions

Recent seismological observations focusing on the collapse of an impulsive wavelet revealed the existence of small-scale random heterogeneities in the earth medium. The RTT has been often used for the study of the propagation of wavelet intensities, the MS amplitude envelopes. For the statistical characterization of the PSDF of random velocity inhomogeneities in a 3D space, we have used von Kármán type with three parameters: the RMS fractional velocity fluctuation $\varepsilon$, the characteristic length $a$, and the order $\kappa$ of the modified Bessel function of the second kind. This model leads to the power-law decay of PSDF $\propto m^{-2\kappa-3}$ at wavenumber $m$ higher than the corner at $a^{-1}$. We have compiled reported statistical parameters of the lithosphere and the mantle based on various types of measurements for a wide range of wavenumbers: photo scan data of rock samples, acoustic well log data, and envelope analyses of cross-hole experiment seismograms, regional seismograms and tele-seismic waves based on the RTT. Reported $\kappa$ values are distributed between 0 and 0.5 (PSDF $\propto m^{-3\sim-4}$), where many of them are close to 0 (PSDF $\propto m^{-3}$). Reported $\varepsilon$ values are of the order of 0.01~0.1 in the crust and the upper mantle, where smaller values in the lower mantle and higher values beneath volcanoes. Reported $a$ values distribute very widely, however, each one seems to be restricted by the dimension of the measurement system or the sample length. In order to grasp the spectral characteristics, eliminating strong heterogeneity data and the lower mantle data, we have plotted all the reported PSDFs in the crust and the upper mantle against wavenumber $m$ for a wide range $10^{-3} \sim 10^8$ km$^{-1}$. We find that the envelope of those PSDFs is well approximated by a power-law decay curve $0.01\,m^{-3}$ km$^3$. Multiple plots of PSDFs and the power-law decay spectral envelope obtained require us to do the followings: In theory, it will be necessary to examine whether the Born approximation is reliable or not if the wavenumber increases much larger than the corner; in observation, we will have to examine more carefully the behavior of each PSDF on both sides of the corner. If we accept the power-law decay spectral envelope, it suggests that the earth medium randomness has a broad spectrum. We may consider the obtained power-law decay spectral envelope as a refer-

ence for studying the regional differences. It is interesting to study what kinds of geophysical processes created the power-law spectral envelope in different scales and in different geological environments of the solid earth medium.

*Competing interests.* The author declares that he has no conflict of interest.

*Acknowledgements.* This paper is a part of the Beno Gutenberg medal lecture at the 2018 EGU assembly held in Vienna. The author is
5  grateful to the seismology section of EGU for giving him an opportunity for reviewing measurements of random heterogeneities in the solid earth and various theoretical approaches. The author expresses sincere thanks to his colleagues and ex-graduate students who enthusiastically collaborated with him in studying seismic wave scattering in random media. The author acknowledges reviewers, Vernon Cormier, Michael Korn and Ludovic Margerin and editor Tarje Nissen-Meyer for their helpful comments and suggestions.

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

**Table 1.** Reported von Kármán parameters of rock samples and acoustic well logs. A value in ( ) is a priori assumed in the measurement. A label with an asterisk * is insufficient for plotting the PSDF.

| Label | Rock type | 1D-PSDF [km] | $\kappa$ | $\varepsilon$ | $a$ [km] | Wavenumber $m$ range [km$^{-1}$] | Reference |
|---|---|---|---|---|---|---|---|
| **Photo scan of rock samples** | | | | | | | |
| R1 | Westerly (fine) Granite, Vp, 1D | – | (0.5) | 0.085 | $0.22 \times 10^{-6}$ | $(1\sim50)\times10^{6}$ | Sivaji et al. (2002) |
| R2 | Oshima (medium) Granite, Vp, 1D | – | (0.5) | 0.093 | $0.46 \times 10^{-6}$ | $(1\sim50)\times10^{6}$ | Sivaji et al. (2002) |
| R3 | Inada (coarse) Granite, Vp, 1D | – | (0.5) | 0.079 | $0.92 \times 10^{-6}$ | $(1\sim50)\times10^{6}$ | Sivaji et al. (2002) |
| R4 | Oshima (medium) Granite, Vs, 1D | – | (0.5) | 0.17 | $0.39 \times 10^{-6}$ | $(0.15\sim10)\times10^{6}$ | Fukushima et al. (2003) |
| R5 | Tamura Gabbro, Vs, 1D | – | (0.5) | 0.081 | $0.84 \times 10^{-6}$ | $(0.15\sim10)\times10^{6}$ | Fukushima et al. (2003) |
| **Acoustic well logs** | | | | | | | |
| W1 | UM crust, YT-1, Japan, Vp, 1D | $3.3 \times 10^{-7}\left(\frac{m}{10^4}\right)^{-1.45}$ | 0.225 | – | – | $(0.02\sim2)\times10^{3}$ | Shiomi et al. (1997) |
| W2 | UM crust, IWT, Japan, Vp, 1D | $3.3 \times 10^{-7}\left(\frac{m}{10^4}\right)^{-1.09}$ | 0.045 | – | – | $(0.01\sim2)\times10^{3}$ | Shiomi et al. (1997) |
| W3* | U crust, KTB, Germany, Vp, 1D | $\propto m^{-0.97}$ | −0.015 | – | $a_z > 1, a_h/a_z = 1.8$ | $(0.001\sim0.5)\times10^{3}$ | Wu et al. (1994) |
| W4 | U crust, KTB, Germany, Vp, 1D | $\propto m^{-1.2}$ | 0.10 | 0.048 | 0.16 | $(0.001\sim0.4)\times10^{3}$ | Holliger (1996) |
| W5 | U crust, Cajon Pass, Ca., Vp, 1D | $\propto m^{-1.22}$ | 0.11 | 0.067 | 0.14 | $(0.001\sim0.4)\times10^{3}$ | Holliger (1996) |

**Table 2.** Reported von Kármán parameters of the lithosphere including the crust and the most-upper mantle. A value in ( ) is a priori assumed in the measurement. When the estimated value is given by a range, a value in [ ] is used as a representative for plotting the PSDF. A label with an asterisk * is insufficient for plotting the PSDF.

| Label | Region | Data & Method | 3D -PSDF [km³] | $\kappa$ | $\varepsilon$ | $a$ [km] | $f$ range [Hz] | $m$ range [km⁻¹] | Reference |
|---|---|---|---|---|---|---|---|---|---|
| **Lithosphere (crust and upper-most mantle)** | | | | | | | | | |
| L1 | M. U. crust, California | Vp Tomography, 3D | – | 0.04 | 0.107 | $a_z = 0.1$ $a_h = 0.51$ | | 1~80 | Nakata and Beroza (2015) |
| L2 | M. U. crust, Michigan | P&S env., Born 2D | – | (0.5) | [0.03] 0.01~0.05 [0.03] | 0.0005~0.002 [0.001] | (1.2~2.8)×10³ | (2~9.4)×10³ | da Silva et al. (2018) |
| L3 | Lithos., Grafenberg | tele.-P env., FD | – | (0.5) | 0.029 | 0.7~16 [1.8] | 1.5~2.5 | 1.3~4.6 | Rothert and Ritter (2000) |
| L4 | Crust, Grafenberg | P env., Born | – | (0.5) | 0.059 | 0.27 | 4~8 | 3.6~14 | Gaebler et al. (2015) |
| L5 | Crust, Norway | P&S env., Born | – | 0.2 | 0.03 | 4 | 2~10 | 3.6~36 | Przybilla et al. (2009) |
| L6 | Crust, Pyrenees | S coda, Born, | – | 3 | 0.021 | 0.09 | 3~12 | 5.4~43 | Calvet and Margerin (2013) |
| L7 | Crust, Pyrenees | P, S & Lg env., Born | – | (0.5) | 0.02 | 0.77 | 2~4 | 3.6~14 | Sens-Schönfelder et al. (2009) |
| L8 | M. U. mantle, Pyrenees | P, S & Lg env., Born | – | (0.5) | 0.05 | 2 | 2~4 | 3.6~14 | Sens-Schönfelder et al. (2009) |
| L9 | Crust, U. mantle, W. Japan | S-env., FD | – | (0.5) | 0.07 | 3.1 | 0.063~0.5 | 0.1~0.7 | Emoto et al. (2017) |
| L10 | Crust, W. Japan. | SH amp., FD | – | (0.5) | 0.03 | 3~5 [4] | 2~8 | 3.6~14.4 | Takemura et al. (2009) |
| L11 | Crust, W. Japan | P env., FD | – | (0.5) | 0.042 | 1 | 0.75~12 | 0.67~11 | Kobayashi et al. (2015) |
| L12 | Lithos., Fore-arc, NE Japan | S peak-delay, Markov | $0.008\,m^{-4.2}$ | 0.6 | 0.07 | [5] | 2~32 | 3.6~57 | Takahashi et al. (2009) |
| L13 | Lithos., Fore-arc, NE Japan | S env. broad., Markov | – | 0.8 | 0.04 | [5] | 2~16 | 3.6~29 | Saito et al. (2005) |
| L14 | Crust, Lop Nor, China | Pg, Lg env., Born | – | 0.3 | 0.008 | 0.2 | 1~4 | 1.8~14 | Sanborn et al. (2017) |
| L15 | M. U. mantle, Lop Nor | Pg, Lg env., Born | – | 0.5 | 0.4 | 0.2 | 1~4 | 1.8~14 | Sanborn et al. (2017) |
| L16 | M. U. crust, California | P, S env., Born | – | (0.3) | 0.05 | 0.05 | 2~4 | 3.6~14 | Wang and Shearer (2017) |
| L17 | L. crust, California | P, S env., Born | – | (0.3) | – | 2 | 2~4 | 3.6~14 | Wang and Shearer (2017) |
| L18* | Lithos., Kamchatka | S env. broad., Born | $\propto m^{-3.85}$ | 0.425 | 0.038~0.043 [0.04] | – | 0.5~16 | 0.9~29 | Petukhin and Gusev (2003) |
| L19 | Lithos., Europe, Group 1 | P env., EFM, TFWM | – | (0.5) | 0.055~0.063 [0.06] | 1.5~4.7 [3] | 0.5~5 | 0.4~4.2 | Hock et al. (2004) |
| L20 | Lithos., Europe, Group 2 | P env., EFM, TFWM | – | (0.5) | | 1.5~2.5 [2] | 0.5~5 | 0.4~4.2 | Hock et al. (2004) |
| **Strong heterogeneities** | | | | | | | | | |
| LS1 | Nikko, Japan | P &S coda, Born | – | (0.5) | 0.063 | 0.5 | 8~16 | 14~59 | Yoshimoto et al. (1997b) |
| LS2 | Kurikoma volc. NE Japan | S peak-delay, Markov | $0.020\,m^{-3.7}$ | 0.35 | 0.061 | [5] | 2~32 | 3.6~57 | Takahashi et al. (2009) |
| LS3 | Taal volc. Philippines | S-env., Born | – | (0.5) | 0.2 | 0.05 | 5~10 | 12~50 | Morioka et al. (2017) |
| LS4 | Pyrenean hetero. body | P, S& Lg env., Born | | (0.5) | 0.072 | 0.77 $a_p=10$, $a_t=0.5$ | 2~4 | 3.6~14 | Sens-Schönfelder et al. (2009) |
| LS5 | Oceanic plate, Japan | S env., FD | – | (0.5) | 0.02 | | 0.25~5 | 0.34~9 | Furumura and Kennett (2005) |
| **Array analysis** | | | | | | | | | |
| LA1 | Crust, U. mantle, Montana | tele.-P, Array | – | (Gaussian) | 0.04 | 10 | 0.5 | 0.3~0.6 | Aki (1973) |
| LA2 | Crust, U. mantle, Montana | tele.-P, Array | – | (Gaussian) | 0.019 | 12 | 0.8 | 0.5~1 | Capon (1974) |
| LA3 | Crust, S. California | tele.-P, Array | – | (Gaussian) | 0.0326 | 25 | 1 | 0.54~1.1 | Powell and Meltzer (1984) |
| LA4 | Crust, U. mantle, Norway | tele.-P, Array | $\propto m^{-4}$+W. N. | < 0.5 | 0.01~0.04 | – | 1~3 | 0.05~1.2 | Flatté and Wu (1988) |

**Table 3.** Reported von Kármán parameters of the upper mantle and the lower mantle. A value in ( ) is a priori assumed in the measurement. When the estimated value is given by a range, a value in [ ] is used as a representative for plotting the PSDF. A label with an asterisk * is insufficient for plotting the PSDF.

| Label | Region | Data & Method | 3D-PSDF [km³] | $\kappa$ | $\varepsilon$ | $a$ [km] | $f$ range [Hz] | $m$ range [km$^{-1}$] | Reference |
|---|---|---|---|---|---|---|---|---|---|
| **Upper mantle** | | | | | | | | | |
| MU1* | U. mantle | Vs Tomography, 2D | $\propto m^{-2\sim-3}\,\mathrm{km}^2$ | 0~0.5 | – | >500 | – | 0.006~0.01 | Chevrot et al. (1998) |
| MU2 | U. mantle | Tele. P env., SEM, Born | – | 0.05 | 0.1 | 2000 | 0.017~0.2 | 0.01~0.14 | Mancinelli et al. (2016a) |
| MU3 | U. mantle | Tele. P env., Born | – | (0.5) | 0.03~0.04 [0.035] | 4 | 0.5~2 | 0.34~1.4 | Shearer and Earle (2004) |
| **Lower mantle** | | | | | | | | | |
| ML1 | L. mantle | Tele. P env., Born | – | (0.5) | 0.005 | 8 | 0.5~2 | 0.26~1 | Shearer and Earle (2004) |
| ML2 | M. L. mantle | PKIKP prec., FD | – | (Gaussian) | 0.01 | 20 | 0.02~2 | 0.01~1 | Cormier (1999) |
| ML3 | M. L. mantle | PKP prec., Born | $2.5 \times 10^{-5}\,m^{-3}$ | 0, HG | 0.002 | $\gg 15$ [30] | 0.5~4 | 0.26~2.1 | Mancinelli et al. (2016b) |
| ML4 | M. L. mantle | PcP env., Born, FD | – | (0.5) | 0.002~0.01[0.005] | 8 | 0.8 ~1.5 | 0.39~0.72 | Zhang et al. (2018) |
| **Whole mantle** | | | | | | | | | |
| MW1 | Whole mantle | PKP prec., Born | $1.4 \times 10^{-5}\,m^{-3}$ | 0, HG | 0.001~0.002 [0.0015] | $\gg 8$ [30] | 0.4~2.5 | 0.2~1.3 | Margerin and Nolet (2003) |
| MW2 | Whole mantle | PP prec., Born | – | (0.5) | 0.008~0.01 [0.009] | 8 | 0.5~2.5 | 0.26~1.3 | Bentham et al. (2017) |