# Peer review of "Power Spectra of Random Heterogeneities of the Solid Earth"

_Solid Earth, 2018_

## Referee Comment (RC1) · Korn (Referee) · 23 Nov 2018

General comments: This is a very informative, compact and comprehensive paper that on the one hand summarizes some of the scientific achievements on seismic wave scattering that Haruo Sato has performed over the years, and at the same time puts many findings on the random heterogeneous structure inside the Earth and the Earth's materials from observations on many scales into a larger common perspective. There is short review of numerical methods to simulate wave envelopes, and their limitations. I like the flowchart figures which illustrate the similarities and differences between approaches without going into much detail. It is remarkable that it seems possible to describe statistical heterogeneity on scales ranging from rock samples to the lower mantle with one unified concept of power spectral density functions with power-law de-

cay at high wavenumbers governed by just three parameters. And there is a hint that maybe even attenuation can be included into that framework to overcome the traditional separation between intrinsic absorption and scattering attenuation. Overall this paper makes pleasant reading and provides a bracket between many observations that can eventually form a step towards uniform description of the Earth's randomness. In this respect the paper not only is a concise short review of past research but also stimulates new ideas.

Specific comments: It could be stated more clearly that the measurements of heterogeneity listed in the Tables are by no means the only ones. Apart from that I don't have specific comments for improvement.

Technical corrections: p.2, L. 24: Correct: When the center wavenumber of a wavelet becomes much larger... p. 2, L. 34: delete "of this paper" p.3 L.8: characterized -> characterize p.3 L. 11: delete superscript of PSDF p. 7 L. 12: typo "directly the " p.8 , L. 14: Two approximations -> The two approximations P. 9 Fig. 5 Typo distorted (2x) p. 10, L. 17: typo through p. 13, L. 2 put L16 and LS3 in brackets p. 13, L. 14: looks a sif an extension -> looks like an extension

---

## Referee Comment (RC2) · Cormier (Referee) · 26 Nov 2018

This is a timely and comprehensive review of the results for 3-D heterogeneity in the crust and mantle obtained from analysis of well-logs, body wave coda modeling, phase fluctuations of observed arrays, velocity tomography, numerical modeling, and radiative transport modeling. Key work on the validity of radiative transport modeling with the Born approximation is cited, calling attention to the use of phase screen approximations in the ma» 1 regime. Figure 8, which summarizes the heterogeneity power spectrum over a broad range of wavenumbers will be a valuable reference for future studies to use for comparison and refinement.

My comments for consideration are related to the validity of common assumptions of

the heterogeneity power spectrum as a function of depth, the interpretation of the spectrum in terms of rheology, temperature, phase, and chemistry, and spectral complexity that may be hidden in log-log plots. These are:

(1) Validity of dlnVp/Vs = 1 assumed in the majority of coda studies. Although not cited in the review, the majority of coda studies employ it to simplify the scattering coefficients. Observationally in the crust and lithosphere it has been measured to be up to 1.5 (Koper).and in the deeper mantle, it has been observed to be > 2 (Romanowicz and others). Thermal effects and viscoelastic attenuation effect have been invoked to explain the observations.

(2) The validity of dlinrho/dlnVs = 0.8. Although the starting point for this assumption has been Birch's law. Even the earliest literature suggested it breaks down with depth. In the deeper mantle, it would lead to very strong buoyancy effects and geodynamic modelers typically assume 0.1 to 0.2 (e.g., Forte et al.)

(3) Validity of (1) and (2) are sufficiently validated in the crust and lithosphere, but it would be important to note the depth of the validation from common assumptions or measurements of lithosphere thickness. 100 to 200 km?

(4) Incorporation of tomography determined heterogeneity. Even accounting for resolution limitations, there frequently has been a discrepancy in the intensity of the true heterogeneity power, measured by velocity fluctuation, epsilon. This is due to the effect of damping required in the inversion. When modelers try to match some observed waveform effects (multi-pathing) starting from tomographic models, they have shown that a factor of 2 or more must be applied to the tomographic inferred velocity fluctuations, e.g., Helmberger, Romanowicz, and co-workers. This scale factor may not be important at the x-y log scales of Figure 8, but still needs to be considered.

(5) All attenuation in the mantle due to scattering. Riccard's suggestion is extreme, but is still important to highlight. It can probably easily be shown to be extreme if one considers the dispersive effect of intrinsic attenuation. The apparent dispersion

of pure scattering attenuation on a body wave pulse will be too small to account for the difference between body wave Earth models versus free oscillation derived free oscillation models, first noted by Dziewonski in the early 1970's. The contribution of scattering attenuation to total attenuation of teleseismic body waves is still an important problem to resolve. The minimum we can say at this point, however, is that the estimate of intrinsic attenuation derived from teleseismic body waves is probably always an over-estimate unless we are able to determine the scattering contribution.

(6) Kolmogorov spectrum. Although viscosity is large, there still may be some validity to consider the shapes and domains of this spectrum for thermally driven convection, similar to its original application to atmospheres. Most of the small-scale heterogeneity in the lithosphere is at scales (a <+ 10km) is most certainly chemical not thermal based on the estimated thermal diffusivities of known mantle materials. This small-scale material is not directly related to a Kolmogorov spectrum, but it is quite possible that larger scales (500 km and greater) are.

(7) Smoothness and complexity of heterogeneity spectrum. The mechanisms creating Earth heterogeneity argue for some complexity that may be hidden at a log-log scale. It is possible that over a broad scale, the heterogeneity spectra is multi-modal in character, with each mode driven by a fundamentally different mechanism. At large scale (several hundred kilometers and greater) there may be a thermally driven mode; at small scale may there is more of a pure chemical signature. For example, at the larger scale Stixrude and Bertolini-Lithgow have predicted peaks in the temperature derivative of upper mantle velocities due to chemical and phase stability at different depths, We (Cormier, Commun. Comp. Phys., accepted) have found a complex signature of lager scale upper heterogeneity that agrees very well with Stixrude and Bertolini-Lithgow. This spectrum consists of intense peaks in epsilon as a function of depth, separated by regions of low epsilon. My hunch is that mantle heterogeneity can be best explained by a superposition of this thermal/chemical large-scale heterogeneity (complex in depth) on top of a small-scale, convectively entrained, chemical heterogeneity.

(8) In the crust, there can also be rheogically driven divisions in a depth dependent small-scale heterogeneity, influenced by brittle-ductile transitions. Some evidence of this has been suggested by Rachman and Chung (BSSA, 2016), Badi et al. (GRL,2009) Bianco et al., (GJI, 2005).

Vernon F. Cormier, University of Connecticut

---

## Author Comment (AC1) · 26 Nov 2018

Dear Prof. Korn,

I am grateful to you for your comments on my review paper.

I first intended to include measurement of intrinsic attenuation in parallel with the measurement of power spectral density function of random velocity heterogeneity; however, I give up to summarize those measurements of intrinsic attenuation since various different assumptions are used in different measurements.

In future, it will be also necessary for us to evaluate scattering contribution of pores, cracks and fractures, which are not considered in the present review work, in addition to the scattering contribution of random heterogeneity.

[Figure]

I will revise the paper reflecting your specific comments on tables and technical corrections.

Thanks again,

Haruo SATO

---

## Author Comment (AC2) · 29 Nov 2018

I am grateful to Prof. Cormier for his informative comments, which are helpful for revising the manuscript.

(1-3) About the validity of d ln Vp/Vs = 1, dlnrho/dlnVs = 0.8, and their depth dependence. . ..

Reply: I agree with the reviewer's comments. I introduced the reduction of the number of independent randomness in order to simplify the mathematics of elastic wave scattering process (Sato, 1984). Those assumptions are appropriate for shallow depths as the reviewer says; however, many papers use these assumptions in their analyses even in the mantle. I will write comments on this problem in the revised manuscript.

[Figure]

(4) Resolution limit of the tomography . . ..

Reply: I agree that PSDFs estimated from tomography have a problem because of their resolutions. PSDFs estimated from tomography are plotted in Figure 3 a, but they are not plotted in Figure 8, where the annotation "Velocity tomography" in the middle of the figure is mistyped. I will correct it in the revision. As enumerated in Tables, tomography results show 0<kappa<0.5, which seems to be reliable.

(5) scattering and intrinsic attenuation. . .

Reply: I first intended to review the measurement of intrinsic attenuation in parallel with the estimate of scattering coefficient. But I give up this task since various different assumptions are used in different measurements. Therefore, I do not discuss about intrinsic attenuation in this review paper.

(6) Kolmogorov spectrum... . Although viscosity is large, there still may be some validity to consider the shapes and domains of this spectrum for thermally driven convection, similar to its original application to atmospheres. . .but it is quite possible that larger scales (500 km and greater) are. . ..

Reply: I also imagine a possibility for the existence of random structure derived from convention motion in the mantle. However, I can not find any literatures which physically derive the power-law spectrum according to the Kolmogorov cascade like process for high viscosity fluids. Therefore, I do not mention about the physical origin and I would like to eave it for future studies.

(7) Smoothness and complexity of heterogeneity spectrum. . .

Reply: I agree that the heterogeneity spectra over a broad scale is multi-modal in character, with each mode driven by a fundamentally different mechanism as the reviewer says. I will refer to Stixrude and Lithgow-Bertelloni (2007) on the velocity variation due to chemical and phase stability at different depths, which is a possible candidate especially for the heterogeneity in the vertical direction.

(8) Depth dependence of small scale heterogeneity...

Reply: I will comment these works based on the MLTW analysis supposing isotropic scattering in the revised version.
* * *

---

## Referee Comment (RC3) · Margerin (Referee) · 3 Dec 2018

The manuscript "Power Spectra of Random Heterogeneities of the Solid Earth" summarizes in an accessible and comprehensive way more than 30 years of measurements and observations of Earth heterogeneity on a very broad range of scales (from $10^{-8}$ to $10^4$ kms). While the focus is put on seismic methods, other approaches (well-logging, direct observations from rock samples) are also presented. Furthermore, the author provides interesting research directions for future seismological developments, in particular he introduces the distorted wave Born approximation for the modeling of energy transport in the high-frequency regime ka » 1, where k is the central wavenumber of the wave and a the correlation distance. In the future, it would be interesting to see how this method may be extended to polarized elastic waves. The paper is well illustrated and

very pleasant to read. It will be a very useful reference for seismologists interested in the stochastic description of Earth Heterogeneity as well as other geoscientists eager to understand how seismologists map small-scale heterogeneities.

General question:

Figure 8 is a central result of the compilation made by the author, where he demonstrates a universal feature of the power spectrum of elastic fluctuations in the Earth: namely that this spectrum is very well described by the von-Karman model with an exponent close to 0, suggesting that the different envelopes of the Earth are rich in small-scales. This Figure is also a source of interrogation. If the power-law is universal then does it contain information on the dynamic processes that are at the origin of the heterogeneity? Indeed, one would expect that different tectonic processes occurring in different envelopes of the Earth leave different imprints in the power spectrum of heterogeneities at small-scale. And to some extent, this is what seismological observations -recalled by the author- also suggest. For example, the observation of guided waves indicate the presence of small-scale heterogeneities with anisotropic scale-lengths in subducting slabs. In Japan, Pulse broadening is wildly different between fore-arc and back-arc regions. Yet Figure 8 seems to imply that the same power spectrum can match completely different geological environments at different scales. Therefore, I wonder whether the important information is really contained in the exponent of the power spectrum or if it should be used in conjunction with other measurements like frequency dependent attenuation, vp/vs ratio, etc. . . or if the model should be complexified (introduction of anisotropic scale lengths)?

Technical questions:

(1) The author rightfully points out that the use of the Born approximation (BA) is problematic at high-frequency. Indeed BA predicts an increase of attenuation without limit as ka tends to infinity. The author also suggests that the limit of applicability of BA is the same as the limit of applicability of the Bourret approximation from mean-field theory.

Yet, the catastrophic increase of attenuation predicted by BA does NOT occur in the Bourret approximation. Bourret approximation does in fact predict the same geometrical limit as the phase screen approximation, although in a much less transparent way since it involves the solution of an implicit equation for the wavenumber of the mean field. It is only when the solution of this implicit equation is simplified by employing the BA that the Bourret approximation fails. But conceptually, I think that the limit of validity of the 2 approximations should be distinguished.

(2) The author explains that the Phase Screen Approximation cannot model the coda. It is not clear to me how the sentence should be interpreted. Certainly the method cannot model wide-angle scattering. At the same time, if the random walker takes a large number of steps, it may eventually come back to its starting point, thereby generating a coda. In optics, this situation is very common. For instance, light diffusion in tissues is in a regime of very strong forward scattering, where the transport mean free path (the length scale of the diffusion process) is much larger than the mean free path (the length of a single step of the random walker). Could the author elaborate a bit on this point?

(3) In the text, the author refers to various estimates of the power spectrum of heterogeneities based on different sampling of the random process. Some estimates are based on 1-D sampling, others on 2-D sampling. It would be interesting to briefly recall how one extrapolates from either a 1-D or 2-D power spectrum to a 3-D one. What are the necessary assumptions (isotropy?) and what is the relation between the measured 1-D or 2-D power law and its 3-D extrapolation?

(4) Would it be possible to explain in simple terms why the classical BA and the phase-screen approximation disagree for ka < 0.2 in the example shown in Figure 5? More generally is there a simple criterion which could be employed to know whether one should employ the BA or its distorted-wave version?

---

## Author Comment (AC3) · 5 Dec 2018

I am grateful to Dr. Margerin for his precise comments. I will revise the paper reflecting his comments and suggestions.

General question: Figure 8 is a central result of the compilation . . .. If the power-law is universal then does it contain information on the dynamic processes that are at the origin of the heterogeneity? Indeed, one would expect that different tectonic processes occurring in different envelopes of the Earth leave different imprints in the power spectrum of heterogeneities at small-scale. And to some extent, this is what seismological observations -recalled by the author- also suggest. For example, the observation of guided waves indicate the presence of small-scale heterogeneities with

anisotropic scale-lengths in subducting slabs. In Japan, Pulse broadening is wildly different between fore-arc and back-arc regions. Yet Figure 8 seems to imply that the same power spectrum can match completely different geological environments at different scales. Therefore, I wonder whether the important information is really contained in the exponent of the power spectrum or if it should be used in conjunction with other measurements like frequency dependent attenuation, vp/vs ratio, etc. . . or if the model should be complexified (introduction of anisotropic scale lengths)?

Reply: Most of kappa values are distributed between 0 and 0.5 and mechanisms which create random heterogeneities are thought to vary in different environments. Figure 8 simply shows that the envelope of various power spectra well obeys a power law with kappa =0 for a wide range of wave-numbers. Not that the gray line is not the average spectrum but the spectral envelope. So far I do not have adequate answer to the above question. But, I still believe it is geophysically important to measure the power spectrum of random heterogeneities in each environment.: the location the corner wavenumber, how the flat level extends into the lower wavenumber, and how the decaying branch extends into higher wavenumbers. It is also necessary to examine the validity of mathematics used in each analysis. We will have to study more about the anisotropy of randomness, intrinsic absorption, and scattering contribution of cracks/pores, which were not considered in this review.

Technical questions: (1) The author rightfully points out that the use of the Born approximation (BA) is problematic at high-frequency. Indeed BA predicts an increase of attenuation without limit as ka tends to infinity. The author also suggests that the limit of applicability of BA is the same as the limit of applicability of the Bourret approximation from mean-field theory. Yet, the catastrophic increase of attenuation predicted by BA does NOT occur in the Bourret approximation. Bourret approximation does in fact predict the same geometrical limit as the phase screen approximation, although in a much less transparent way since it involves the solution of an implicit equation for the wavenumber of the mean field. It is only when the solution of this implicit equation is

simplified by employing the BA that the Bourret approximation fails. But conceptually, I think that the limit of validity of the 2 approximations should be distinguished.

Reply: Thank you for the above comments on the Bourret approximation. I will restrict myself to point out the problem of the conventional Born app. in this paper. I will delete the sentence about the Bourret approximation on page 8 ". . . . . . which is the same as the criterion of the Bourret approximation (Rytov et al., 1989)." in the revision.

(2) The author explains that the Phase Screen Approximation cannot model the coda. It is not clear to me how the sentence should be interpreted. Certainly the method cannot model wide-angle scattering. At the same time, if the random walker takes a large number of steps, it may eventually come back to its starting point, thereby generating a coda. In optics, this situation is very common. For instance, light diffusion in tissues is in a regime of very strong forward scattering, where the transport mean free path (the length scale of the diffusion process) is much larger than the mean free path (the length of a single step of the random walker). Could the author elaborate a bit on this point?

Reply: I understand that you are talking about the coherent back scattering phenomena. I agree with the reviewer's comment. As shown in Figure 8 of Sato and Emoto (GJI, 2018), a decaying coda is shown according to the RTT with the phase screen app. In the revised version, I will rephrase this as follows: " The phase screen approximation is not adequate for wide angle scattering but for narrow angle scattering."

(3) In the text, the author refers to various estimates of the power spectrum of heterogeneities based on different sampling of the random process. Some estimates are based on 1-D sampling, others on 2-D sampling. It would be interesting to briefly recall how one extrapolates from either a 1-D or 2-D power spectrum to a 3-D one. What are the necessary assumptions (isotropy?) and what is the relation between the measured 1-D or 2-D power law and its 3-D extrapolation?

Reply: I assume isotropic randomness in the conversion from 1D to 3D. For von Kar-

man type, kappa value is common to different dimensions. Attached pdf shows the mathematics (after Sato et al. 2012).

(4) Would it be possible to explain in simple terms why the classical BA and the phase screen approximation disagree for ka < 0.2 in the example shown in Figure 5?

Reply: In the case k a <1, the conventional Born app. is applicable but the phase screen app. is not appropriate since it is based on the parabolic app.

More generally is there a simple criterion which could be employed to know whether one should employ the BA or its distorted-wave version?

Reply: The criterion epsˆ2 aˆ2 kcˆ2 «O(1) at the bottom of page 8 is a kind of extrapolation from the deterministic scattering case. So far I cannot say the criterion in a simple manner.

**Evaluation of PSDF in 1-D from that in 3-D**    When 3-D random media are isotropic, we can evaluate the PSDF along a line by taking samples along the $z$-axis at $x = y = 0$:

$$
\begin{aligned}
P_{1D}(m_z) &\equiv \int_{-\infty}^{\infty} R_{3D}(0,0,z)\, e^{-im_z z} dz \\
&= \int_{-\infty}^{\infty} \left[ \frac{1}{(2\pi)^3} \iiint_{-\infty}^{\infty} P_{3D}(\mathbf{m}')\, e^{im_z' z} d\mathbf{m}' \right] e^{-im_z z} dz \\
&= \frac{1}{(2\pi)^2} \iint_{-\infty}^{\infty} P(m_x', m_y', m_z)\, dm_x' dm_y'.
\end{aligned}
\tag{1}
$$

In the case of isotropic von Kármán type,

$$
\begin{aligned}
P_{1D}(m_z) &= \frac{1}{(2\pi)^2} \iint_{-\infty}^{\infty} \frac{8\pi^{3/2}\varepsilon^2 a^3 \Gamma(\kappa+3/2)}{\Gamma(\kappa)\left[1 + a^2\left(m_x'^2 + m_y'^2 + m_z^2\right)\right]^{\kappa+3/2}} dm_x' dm_y' \\
&= \frac{2\pi^{1/2}\Gamma(\kappa+1/2)\,\varepsilon^2 a}{\Gamma(\kappa)\left(1 + a^2 m_z^2\right)^{\kappa+1/2}}.
\end{aligned}
\tag{2}
$$

This relation between different dimensions is a key for the interpretation of well logs, which are 1-D sample data.

**Fig. 1.**

---

## Author Response (AR1)

Re: "Power Spectra of Random Heterogeneities of the Solid Earth" by Haruo SATO

Dear Dr. Tarje Nissen-Meyer,
Handling editor of Solid Earth,

I am grateful to three reviewers for their valuable comments which are helpful for revising the manuscript.

I newly added date sets L19 and L20 in Table 2, and corrected several typos in tables.
I have redrawn all the figures to improve the visibility.  I have throughly revised the discussion section, in which new sentences are added.

I am sending two files: one marked-up file uses red color fonts for newly added sentences and texts and deleted texts in the first version; another is the revised version using black font color.

Hope the revised manuscript will fulfill  the review criterion.

Best regards,

Haruo SATO

_______________-

Review of Dr. Korn:
General comments: This is a very informative, compact and comprehensive paper that on the one hand summarizes some of the scientific achievements on seismic wave scattering that Haruo Sato has performed over the years, and at the same time puts many findings on the random heterogeneous structure inside the Earth and the Earth's materials from observations on many scales into a larger common perspective. There is short review of numerical methods to simulate wave envelopes, and their limitations. I like the flowchart figures which illustrate the similarities and differences between approaches without going into much detail. It is remarkable that it seems possible to describe statistical heterogeneity on scales ranging from rock samples to the lower mantle with one unified concept of power spectral density functions with power-law decay  at high wavenumbers governed by just three parameters. And there is a hint that maybe even attenuation can be included into that framework to overcome the traditional separation between intrinsic absorption and scattering attenuation. Overall this paper makes pleasant reading and provides a bracket between many observations that can eventually form a step towards uniform description of the Earth's randomness. In this respect the paper not only is a concise short review of past research but also stimulates new ideas.

Specific comments: It could be stated more clearly that the measurements of heterogeneity listed in the Tables are by no means the only ones. Apart from that I don't have specific comments for improvement.

**Reply >  On P5L15…I added a sentence "Note that the measurements of heterogeneity listed in the Tables are by no means the only ones."**

Technical corrections: p.2, L. 24: Correct: When the center wavenumber of a wavelet becomes much larger. . . p. 2, L. 34: delete "of this paper" p.3 L.8: characterized -> characterize p.3 L. 11: delete superscript of PSDF p. 7 L. 12: typo "directly the " p.8 , L. 14: Two approximations -> The two approximations P. 9 Fig. 5 Typo distorted (2x) p. 10, L. 17: typo through p. 13, L. 2 put L16 and LS3 in brackets p. 13, L. 14: looks a sif an extension -> looks like an extension

**Reply > Corrected.**
**Comment "p. 13, L. 2 put L16 and LS3 in brackets " is left as it is since the instruction is not clear.**
* * *
Review of Dr. Cormier:
This is a timely and comprehensive review of the results for 3-D heterogeneity in the crust and mantle obtained from analysis of well-logs, body wave coda modeling, phase fluctuations of observed arrays, velocity tomography, numerical modeling, and radiative transport modeling. Key work on the validity of radiative transport modeling with the Born approximation is cited, calling attention to the use of phase screen approximations in the ma» 1 regime. Figure 8, which summarizes the heterogeneity power spectrum over a broad range of wavenumbers will be a valuable reference for future studies to use for comparison and refinement. My comments for consideration are related to the validity of common assumptions of the heterogeneity power spectrum as a function of depth, the interpretation of the spectrum in terms of rheology, temperature, phase, and chemistry, and spectral complexity that may be hidden in log-log plots. These are:
(1) Validity of dlnVp/Vs = 1 assumed in the majority of coda studies. Although not cited in the review, the majority of coda studies employ it to simplify the scattering coefficients. Observationally in the crust and lithosphere it has been measured to be up to 1.5 (Koper).and in the deeper mantle, it has been observed to be > 2 (Romanowicz and others). Thermal effects and viscoelastic attenuation effect have been invoked to explain the observations.
(2) The validity of dlinrho/dlnVs = 0.8. Although the starting point for this assumption has been Birch's law. Even the earliest literature suggested it breaks down with depth. In the deeper mantle, it would lead to very strong buoyancy effects and geodynamic modelers typically assume 0.1 to 0.2 (e.g., Forte et al.)
(3) Validity of (1) and (2) are sufficiently validated in the crust and lithosphere, but it would be important to note the depth of the validation from common assumptions or measurements of lithosphere thickness. 100 to 200 km?

**Reply to (1) -(3) >**
**P15L7-14: New sentences reflecting the above comments are added.**

(4) Incorporation of tomography determined heterogeneity. Even accounting for resolution limitations, there frequently has been a discrepancy in the intensity of the true heterogeneity power, measured by velocity fluctuation, epsilon. This is due to the effect of damping required in the inversion. When modelers try to match some observed waveform effects (multi-pathing) starting from tomographic models, they have shown that a factor of 2 or more must be applied to the tomographic inferred velocity fluctuations, e.g., Helmberger, Romanowicz, and co-workers. This scale factor may not be important at the x-y log scales of Figure 8, but still needs to be considered.

**Reply > I agree the above criticism.**

**P7L15, I added one sentence "… there is a resolution limit of the tomography method ".**

(5) All attenuation in the mantle due to scattering. Riccard's suggestion is extreme, but is still important to highlight. It can probably easily be shown to be extreme if one considers the dispersive effect of intrinsic attenuation. The apparent dispersion of pure scattering attenuation on a body wave pulse will be too small to account for the difference between body wave Earth models versus free oscillation derived free oscillation models, first noted by Dziewonski in the early 1970's. The contribution of scattering attenuation to total attenuation of teleseismic body waves is still an important problem to resolve. The minimum we can say at this point, however, is that the estimate of intrinsic attenuation derived from teleseismic body waves is probably always an overestimate unless we are able to determine the scattering contribution.

**Reply > I deleted Riccards work onP17L15.**
**There might be some bias for the estimated PSDF of random velocity inhomogeneity depending on how intrinsic attenuation is assumed in the analysis. In this review, we discard intrinsic attenuation parameters a priori assumed or measured in each paper. I added a sentence for the necessity of intrinsic attenuation in future…**

**P15L4-6:"Although most of measurements used in this review analyzed intrinsic attenuation; however, we did not enumerate them in this review since different assumptions were used in different measurements. It will be necessary for us to measure systematically the PSDF of random heterogeneity in conjunction with intrinsic attenuation.**

(6) Kolmogorov spectrum. Although viscosity is large, there still may be some validity to consider the shapes and domains of this spectrum for thermally driven convection, similar to its original application to atmospheres. Most of the small-scale heterogeneity in the lithosphere is at scales (a <+ 10km) is most certainly chemical not thermal based on the estimated thermal diffusivities of known mantle materials. This small-scale material is not directly related to a Kolmogorov spectrum, but it is quite possible that larger scales (500 km and greater) are.

**Reply > As far as I know, there is no appropriate reference about the Kolmogorov cascade in highly viscous mantle fluid. I mention "For random heterogeneities in the mantle, we imagine complex mantle flow." on P17L12.**

**P17L7-8: I slightly modified the sentence as "However, it may be difficult to apply this cascade model to the mantle since the viscosity of mantle fluid is thought to be high. "**

(7) Smoothness and complexity of heterogeneity spectrum. The mechanisms creating Earth heterogeneity argue for some complexity that may be hidden at a log-log scale. It is possible that over a broad scale, the heterogeneity spectra is multi-modal in character, with each mode driven by a fundamentally different mechanism. At large scale (several hundred kilometers and greater) there may be a thermally driven mode; at small scale may there is more of a pure chemical signature. For example, at the larger scale Stixrude and Bertolini-Lithgow have predicted peaks in the temperature derivative of upper mantle velocities due to chemical and phase stability at different depths, We (Cormier, Commun. Comp. Phys., accepted) have found a complex signature of lager scale upper

heterogeneity that agrees very well with Stixrude and Bertolini-Lithgow. This spectrum consists of intense peaks in epsilon as a function of depth, separated by regions of low epsilon. My hunch is that mantle heterogeneity can be best explained by a superposition of this thermal/chemical large-scale heterogeneity (complex in depth) on top of a small-scale, convectively entrained, chemical heterogeneity.

**Reply> Thanks for suggestions. I added following sentences in discussion:**

**P17L16-20: "Stixrude and Lithgow-Bertelloni (2007) proposed the velocity variation due to chemical and phase stability at different depths, which is a possible candidate especially for the heterogeneity in the vertical direction. If we accept the power-law decay spectrum even at high wavenumbers and in the curst, we will have to study what kinds of geophysical mechanisms created such random medium spectra in different scales and in different environments of temperature and pressure in the solid earth. "**

(8) In the crust, there can also be rheogically driven divisions in a depth dependent small-scale heterogeneity, influenced by brittle-ductile transitions. Some evidence of this has been suggested by Rachman and Chung (BSSA, 2016), Badi et al. (GRL,2009) Bianco et al., (GJI, 2005).

**Reply> Thanks for suggestions. I added following sentences in discussion:**

**P17L21-29: "In advance to the measurements based on the RTT for anisotropic scattering presented here, there have been many measure- ments of the isotropic scattering coefficient $g_{iso}$ in the world on the basis of the RTT with the isotropic scattering assumption (e.g. Sato et al., 2012; Yoshimoto and Jin, 2008). The isotropic scattering model is mathematically tractable, and the multiple lapse-time window analysis (Fehler et al., 1992; Hoshiba, 1993) has been often used for practical analyses. This method essen- tially estimates $g_{iso}$ from the ratio of late coda excitation to the radiated energy irrespective of the envelope broadening. Recent measurements show that $g_{iso}$ decreases with depth (e.g. Rachman and Chung, 2016; Badi et al., 2009). It will be interesting to plot the frequency dependence of reported $g_{iso}$ for a wide range of frequencies and to study the relation with the obtained power spectral envelope shown in Figure 8."**

———————

———————
Review of Dr. Margerin:
The manuscript "Power Spectra of Random Heterogeneities of the Solid Earth" summarizesi n an accessible and comprehensive way more than 30 years of measurements and observations of Earth heterogeneity on a very broad range of scales (from 10^-8 to 10^4 kms). While the focus is put on seismic methods, other approaches (well-logging, direct observations from rock samples) are also presented. Furthermore, the author provides interesting research directions for future seismological developments, in particular he introduces the distorted wave Born approximation for the modeling of energy transport in the high-frequency regime ka » 1, where k is the central wavenumber of the wave and a the correlation distance. In the future, it would be interesting to see how this method may be extended to polarized elastic waves. The paper is well illustrated and very pleasant to read. It will be a very useful reference for seismologists interested in the stochastic description of Earth

Heterogeneity as well as other geoscientists eager to understand how seismologists map small-scale heterogeneities.

Reply> P15L28: I added "It would be interesting to see how this method may be extended to polarized elastic waves.

General question:
Figure 8 is a central result of the compilation made by the author, where he demonstrates a universal feature of the power spectrum of elastic fluctuations in the Earth: namely that this spectrum is very well described by the von-Karman model with an exponent close to 0, suggesting that the different envelopes of the Earth are rich in small-scales. This Figure is also a source of interrogation. If the power-law is universal then does it contain information on the dynamic processes that are at the origin of the heterogeneity? Indeed, one would expect that different tectonic processes occurring in different envelopes of the Earth leave different imprints in the power spectrum of heterogeneities at small-scale. And to some extent, this is what seismological observations -recalled by the author- also suggest. For example, the observation of guided waves indicate the presence of small-scale heterogeneities with anisotropic scale-lengths in subducting slabs. In Japan, Pulse broadening is wildly different between fore-arc and back-arc regions. Yet Figure 8 seems to imply that the same power spectrum can match completely different geological environments at different scales. Therefore, I wonder whether the important information is really contained in the exponent of the power spectrum or if it should be used in conjunction with other measurements like frequency dependent attenuation, vp/vs ratio, etc. . . or if the model should be complexified (introduction of anisotropic scale lengths)?

**Reply > It is difficult to answer the above question. I can say that the power spectral envelope is simply given by a power law curve; however, as shown in tables, most of kappa values distribute between 0 and 0.5.   I personally think the exponent of wavenumber contains fruitful information of the geophysical generation process of the random structure in different environments. I feel there are some ambiguity in  measurements of the correlation length. The use of the conventional Born approximation may contain some difficulty at high wave-numbers.  Thus there are several problems in both measurements and theory.  We will have to study more about the anisotropy of randomness, intrinsic absorption, and scattering contribution of cracks/pores, which were not considered in this review.**

**I added sentences concerning the above points in the revised discussion section.**

**P15L4-6: on intrinsic attenuation.**

**P15L7-13: On the velocity fluctuation.**

**P15L14-17: On the anisotropic random structure.**

**P16L5-6: On scattering by pores and cracks**

Technical questions:

(1) The author rightfully points out that the use of the Born approximation (BA) is problematic at high-frequency. Indeed BA predicts an increase of attenuation without limit as ka tends to infinity. The author also suggests that the limit of applicability of BA is the same as the limit of applicability of the Bourret approximation from mean-field theory. Yet, the catastrophic increase of attenuation predicted by BA does NOT occur in the Bourret approximation. Bourret approximation does in fact predict the same geometrical limit as the phase screen approximation, although in a much less transparent way since it involves the solution of an implicit equation for the wavenumber of the mean field. It is only when the solution of this implicit equation is simplified by employing the BA that the Bourret approximation fails. But conceptually, I think that the limit of validity of the 2 approximations should be distinguished.

**Reply > Thanks for valuable comment. I would like to restrict on the use of the conventional Born approximation. I discard the sentence "which is the same as the criterion of the Bourret approximation." onP9L22.**

(2) The author explains that the Phase Screen Approximation cannot model the coda. It is not clear to me how the sentence should be interpreted. Certainly the method cannot model wide-angle scattering. At the same time, if the random walker takes a large number of steps, it may eventually come back to its starting point, thereby generating a coda. In optics, this situation is very common. For instance, light diffusion in tissues is in a regime of very strong forward scattering, where the transport mean free path (the length scale of the diffusion process) is much larger than the mean free path (the length of a single step of the random walker). Could the author elaborate a bit on this point?

**Reply> I agree with the reviewer's comment. I understand that the reviewer is talking about the coherent back scattering phenomena. As shown in Figure 8 of Sato and Emoto (GJI, 2018), a decaying coda is shown according to the RTT with the phase screen app. In the revised version, I will rephrase this as follows:**
**P13L6-8: "This approximation well synthesizes the intensity time trace having a delayed peak from the onset and a decaying tail of early coda at large travel distances. This approximation can not synthesize the late coda intensity since wide angle scattering is neglected. "**

(3) In the text, the author refers to various estimates of the power spectrum of heterogeneities based on different sampling of the random process. Some estimates are based on 1-D sampling, others on 2-D sampling. It would be interesting to briefly recall how one extrapolates from either a 1-D or 2-D power spectrum to a 3-D one. What are the necessary assumptions (isotropy?) and what is the relation between the measured 1-D or 2-D power law and its 3-D extrapolation?

**Reply> I assume isotropic randomness in the conversion from 1D to 3D. For von Karman type, kappa value is common to different dimensions. Attached pdf shows the mathematics (after Sato et al. 2012). I added math equations (6a) and (6b) on P6L1-8.**

(4) Would it be possible to explain in simple terms why the classical BA and the phase screen approximation disagree for ka < 0.2 in the example shown in Figure 5?

**Reply>  In the case k a <1, the conventional Born app. is applicable but the phase screen app. is not appropriate since it is based on the parabolic app. I added a sentence.**
**P9L18-19: "Note that the phase shift approximation is not applicable for $ak_c < 1$ since it is based on the parabolic approximation."**

More generally is there a simple criterion which could be employed to know whether one should employ the BA or its distorted-wave version?

**Reply>  The criterion eps^2 a^2 kc^2 <<O(1) on P9L21 is a kind of  extrapolation from the deterministic  scattering case. So far I cannot say the criterion in a simple manner.**
* * *

[revised manuscript text omitted]